# Dexamethasone-loaded platelet-inspired nanoparticles improve intracortical microelectrode recording performance

Longshun Li [1,2], Aniya Hartzler[1], Dhariyat M. Menendez-Lustri[1,2], Jichu Zhang[1], Alex Chen[1], Danny V. Lam [1,2], Baylee Traylor[3], Emma Quill[3], David E. Nethery[1,2], George F. Hoeferlin[1,2], Christa L. Pawlowski[3], Michael A. Bruckman[3], Anirban Sen Gupta [1], Jeffrey R. Capadona [1,2] & Andrew J. Shoffstall [1,2] ✉

Long-term robust intracortical microelectrode (IME) neural recording quality is negatively affected by the neuroinflammatory response following microelectrode insertion. This adversely impacts brain-machine interface (BMI) performance for patients with neurological disorders or amputations. Recent studies suggest that the leakage of blood-brain barrier (BBB) and microhemorrhage caused by IME insertions contribute to increased neuroinflammation and reduced neural recording performance. Here, we evaluated dexamethasone sodium phosphate-loaded platelet-inspired nanoparticles (DEXSPPIN) to simultaneously augment local hemostasis and serve as an implant-site targeted drug-delivery vehicle. Weekly systemic treatment or control therapy was provided to rats for 8 weeks following IME implantation, while evaluating extracellular single-unit recording performance. End-point immunohistochemistry was performed to further assess the local tissue response to the IMEs. Treatment with DEXSPPIN significantly increased the recording capabilities of IMEs compared to controls over the 8-week observation period. Immunohistochemical analyses of neuron density, activated microglia/macrophage density, astrocyte density, and BBB permeability suggested that the improved neural recording performance may be attributed to reduced neuron degeneration and neuroinflammation. Overall, we found that DEXSPPIN treatment promoted an anti-inflammatory environment that improved neuronal density and enhanced IME recording performance.

Intracortical microelectrodes (IMEs) are designed to record neuronal activity for neuroscience research and the treatment of neurological disorders[1–4]. Neural signals acquired by IMEs can be analyzed and decoded for brain-machine interface (BMI) applications, enabling control of external devices, prosthetics, and stimulators to restore motor function in patients with limb loss or spinal cord injuries[5–10]. However, the primary obstacle to the clinical translation of BMI technology is the poor long-term reliability and stability of IMEs[11–17].

A key contributor to this decline in performance is the brain's neuroinflammatory response following IME insertion[4,18,19]. Insertion of IMEs into the motor cortex disrupts healthy neurons and the BBB, leading to an acute decline in recording capability and the influx of blood-derived factors into the brain parenchyma[13,15,18,20–27]. These

[1]Department of Biomedical Engineering, Case Western Reserve University, Cleveland, OH, USA. [2]Advanced Platform Technology Center, Louis Stokes Cleveland Department of Veterans Affairs Medical Center, Cleveland, OH, USA. [3]Haima Therapeutics LLC, Cleveland, OH, USA. ✉e-mail: ajs215@case.edu

events trigger neuronal death and the activation and recruitment of resident microglia, which release proinflammatory cytokines and cytotoxic factors. Subsequent recruitment of astrocytes contributes to secondary neuron loss[28–31]. This cascade perpetuates inflammation through damage-associated molecular pattern (DAMP) signaling, ultimately resulting in a dense glial encapsulation of the IME and further degradation of recording quality[32–36]. Additionally, activated platelets and coagulation factors such as von Willebrand Factor (vWF), collagen, and fibrinogen have been observed near the IME interface, suggesting a role in prolonged neuroinflammation, even up to at least 8 weeks post-implantation[37].

To mitigate this response, several groups have tested anti-inflammatory or anti-oxidative strategies including systemic drug delivery, localized administration, and electrode modifications such as coatings or retrodialysis[16,17,28,29,38–44]. For instance, Gaire et al. showed that systemic administration of dexamethasone (DEX) reduced neuroinflammation but did not improve IME recording quality[38]. Moreover, chronic high-dose systemic DEX is contraindicated due risks including bone loss, glucose dysregulation, and organ toxicity[45]. Local DEX delivery via microelectrode coatings, as demonstrated by Zhong et al., improved IME recording outcomes but is limited by single-use release and complicates device commercialization and regulatory approval[39]. These limitations highlight the need for a local drug delivery strategy that does not require modification of existing IME hardware.

Platelet-inspired nanoparticles (PINs) offer a promising solution. They have been shown to colocalize with markers of BBB disruption and platelet activation, thereby localizing to sites of vascular injury near IMEs[46]. PINs bind to exposed collagen and vWF via their collagen- and vWF-binding peptides (CBP and VBP), and to activated platelet GPIIb-IIIa receptors via fibrinogen-mimetic peptides (FMP), facilitating accumulation at the site of vascular injury[47–49]. We hypothesized that loading PINs with dexamethasone sodium phosphate (DEXSP) would enable anti-inflammatory drug delivery to IME implant sites. This approach not only reduces neuroinflammation but also promotes BBB resealing and hemostasis.

Here, we evaluated the therapeutic efficacy of systemically administered DEXSP loaded PINs (DEXSPPIN) in improving long-term IME performance. Using a lipid thin-film rehydration method, we developed a DEXSPPIN formulation that reliably achieves high drug encapsulation efficiency. We then assessed neural recording performance over 8 weeks of weekly treatment via extracellular single-unit recordings. To probe the neurobiological effects of DEXSPPIN, we quantified neuron density (NeuN), microglia activation (CD68) and astrocyte reactivity (GFAP) via immunohistochemistry around the implant site at the study endpoint. Blood-brain barrier integrity was assessed via immunoglobulin-G (IgG) staining. Finally, we evaluated systemic safety by monitoring weight, glucose (GLU), alanine transaminase (ALT) and creatinine (CREA) levels throughout the treatment period.

## Results

### Characterization of DEXSPPIN and DEXSP release profile
As this study represents the first synthesis of dexamethasone sodium phosphate-loaded platelet-inspired nanoparticles (DEXSPPIN), we began by thoroughly characterizing the formulation's encapsulation efficiency, key physicochemical properties, and drug release profile. A schematic of the nanoparticle design is shown in Fig. 1A. Given the hydrophilicity of DEXSP, the drug is presumed to be predominantly loaded within the aqueous core of the particle.

The formulation achieved a high encapsulation efficiency of 78.7 ± 5.5% across seven independently manufactured batches (Fig. 1B). The resulting DEXSPPIN exhibited a surface charge of −13.9 ± 2.6 mV and a mean hydrodynamic diameter of 124.9 ± 16.5 nm, with 90% of particles measuring less than 151.4 ± 26.6 nm (Fig. 1C).

To evaluate drug release kinetics, we used an infinite sink model with dialysis in 1× PBS solution. Approximately 65% of the encapsulated DEXSP was released within the first 24 hours, followed by near-complete release (~98%) over 24 days (Fig. 1D). The biphasic release profile, characterized by an initial burst followed by a sustained phase, informed our decision to prepare fresh DEXSPPIN prior to each weekly administration in the in vivo study.

### Therapeutic efficacy of DEXSPPIN on IMEs recording performance
To evaluate the therapeutic impact of DEXSPPIN on awake recording performance, we implanted 16-channel, single-shank functional IMEs into the primary motor cortex of rats and monitored neural recordings over an 8-week period. Animals were randomly assigned to one of four treatment groups: DEXSPPIN (n = 7), PIN (n = 7), free DEXSP (n = 8), or vehicle control (DILUENT; n = 7). See the timeline in the Methods section.

To quantify recording performance, we measured active electrode yield (AEY) biweekly, defined as the percentage of channels detecting single units (green dots in Fig. 2A) relative to the total viable channels (excluding persistently silent electrodes; red dots in Fig. 2A). AEY serves as a practical measure of electrode function and signal integrity over time.

DEXSPPIN-treated animals demonstrated superior AEY values across the 8-week period, with significantly higher values than both the Free DEXSP and DILUENT groups (Fig. 2B, C). When compared to the PIN group, DEXSPPIN showed significantly improved AEY at most timepoints, except during weeks 3, 4, and 6, suggesting that incorporating targeted drug-delivery of DEXSP contributed positively to functional preservation of neural recordings.

To better understand the temporal trends in recording stability, we grouped data into early (weeks 1-4; W1_4) and late (weeks 5-8; W5_8) study phases (Fig. 2D). In both phases, the DEXSPPIN group maintained a significantly higher proportion of active electrodes compared to all other groups (W1_4: p < 0.001 vs. PIN, p < 0.0001 vs DILUENT Free DEXSP; W5_8: p < 0.0001 vs. all). The PIN group also out-performed the Free DEXSP and DILUENT controls in both phases, highlighting a modest benefit from the targeted nanoparticle platform alone (W1_4: p < 0.05 vs. DILUENT, p < 0.001 vs. Free DEXSP; W5_8: p < 0.001 vs. DILUENT, p < 0.0001 vs. Free DEXSP).

Interestingly, while no significant differences were observed between the DILUENT and Free DEXSP in the first week of the study, by weeks 5-8, the Free DEXSP group exhibited a significantly lower AEY than the DILUENT group (p < 0.001). Within each treatment group, the proportion of active electrodes significantly decreased in the last four weeks of the study (i.e., W5_8 vs W1_4: p < 0.001 in DEXSPPIN, p < 0.0001 in PIN, p < 0.0001 in DILUENT, p < 0.0001 in Free DEXSP), indicating a decline in IME recording performance over time (Fig. S1A–D). Nevertheless, the DEXSPPIN group exhibited a 17% decline in active electrode yield, which was less pronounced compared to the declines in the other groups: 25% for PIN, 37% for DILUENT, and 54% for Free DEXSP. These findings indicate that DEXSPPIN treatment delayed the progressive signal degradation typically seen in chronic IME recordings.

To further assess signal quality, we quantified active single units per channel, reflecting the IMEs' ability to isolate individual neuronal signals (Fig. 2E). During the early phase (W1_4), DEXSPPIN treated animals showed significantly more active units per channel compared to all other groups (p < 0.0001 vs. PIN and Free DEXSP; p < 0.05 vs. DILUENT). In the later phase (W5_8), this advantage persisted over DILUENT (p < 0.05) and DILUENT (p < 0.0001), though no significant difference was detected compared to the PIN group. Within each treatment group, no significant differences were observed between

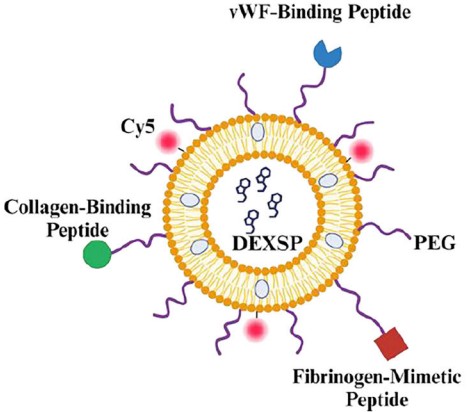

**B**

| Encapsulation Efficiency (%) | Zeta Potential (mV) | Hydrodynamic Diameter (nm) | D90 (nm) |
|---|---|---|---|
| 78.7 ± 5.5 | -13.9 ± 2.6 | 124.9 ± 16.5 | 151.4 ± 26.6 |

**C**

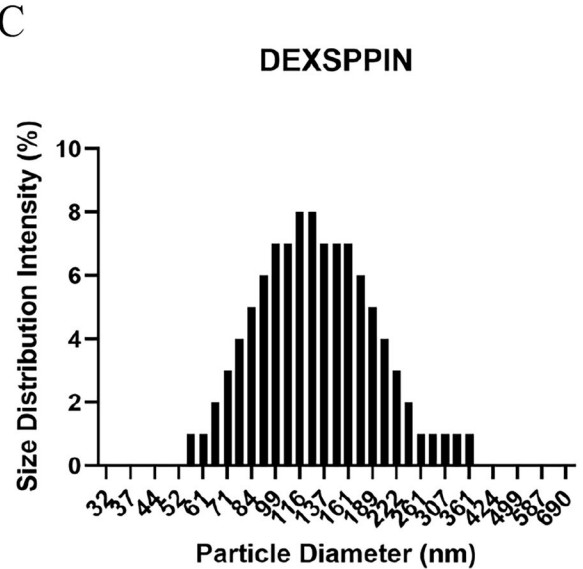

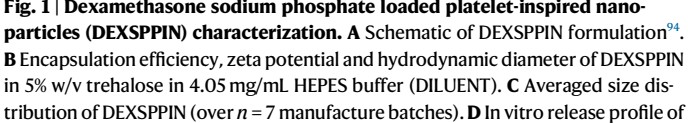

**D**

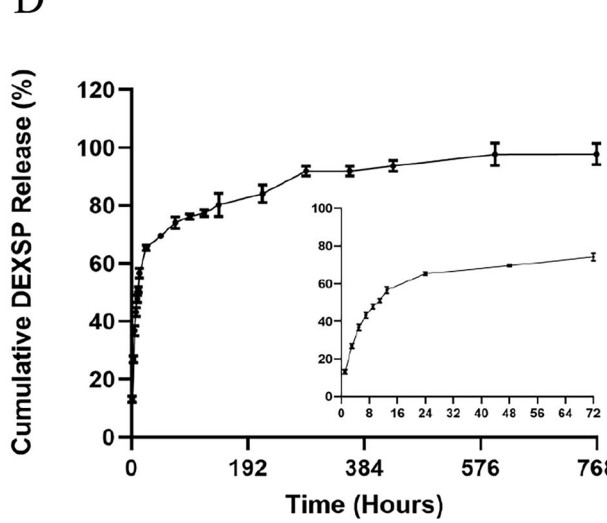

**Fig. 1 | Dexamethasone sodium phosphate loaded platelet-inspired nanoparticles (DEXSPPIN) characterization. A** Schematic of DEXSPPIN formulation[94]. **B** Encapsulation efficiency, zeta potential and hydrodynamic diameter of DEXSPPIN in 5% w/v trehalose in 4.05 mg/mL HEPES buffer (DILUENT). **C** Averaged size distribution of DEXSPPIN (over $n = 7$ manufacture batches). **D** In vitro release profile of dexamethasone sodium phosphate (DEXSP) from platelet-inspired nanoparticles (PIN) into 1 × PBS solution, showing ~ 65% burst release in 24 hr and ~ 98% release in 24 days ($n = 3$, technical replicates). The standard error of the mean (SEM) was used to plot error bars.

the different phases of the study for any treatment group, suggesting that while overall electrode activity declined, signal richness on active channels was preserved.

To assess the quality of recorded signals, we analyzed several parameters including peak-to-peak voltage (Vpp), noise, signal-to-noise ratio (SNR), and spike rate (Fig. 3A–D).

During the first four weeks (W1_4), the DEXSPPIN group exhibited significantly higher Vpp compared to the PIN and Free DEXSP groups ($p < 0.01$), indicating better signal amplitude early in the study (Fig. 3A). However, no significant difference was observed between the DEXSPPIN and DILUENT groups, suggesting that DEXSPPIN treatment may not meaningfully enhance signal strength beyond baseline.

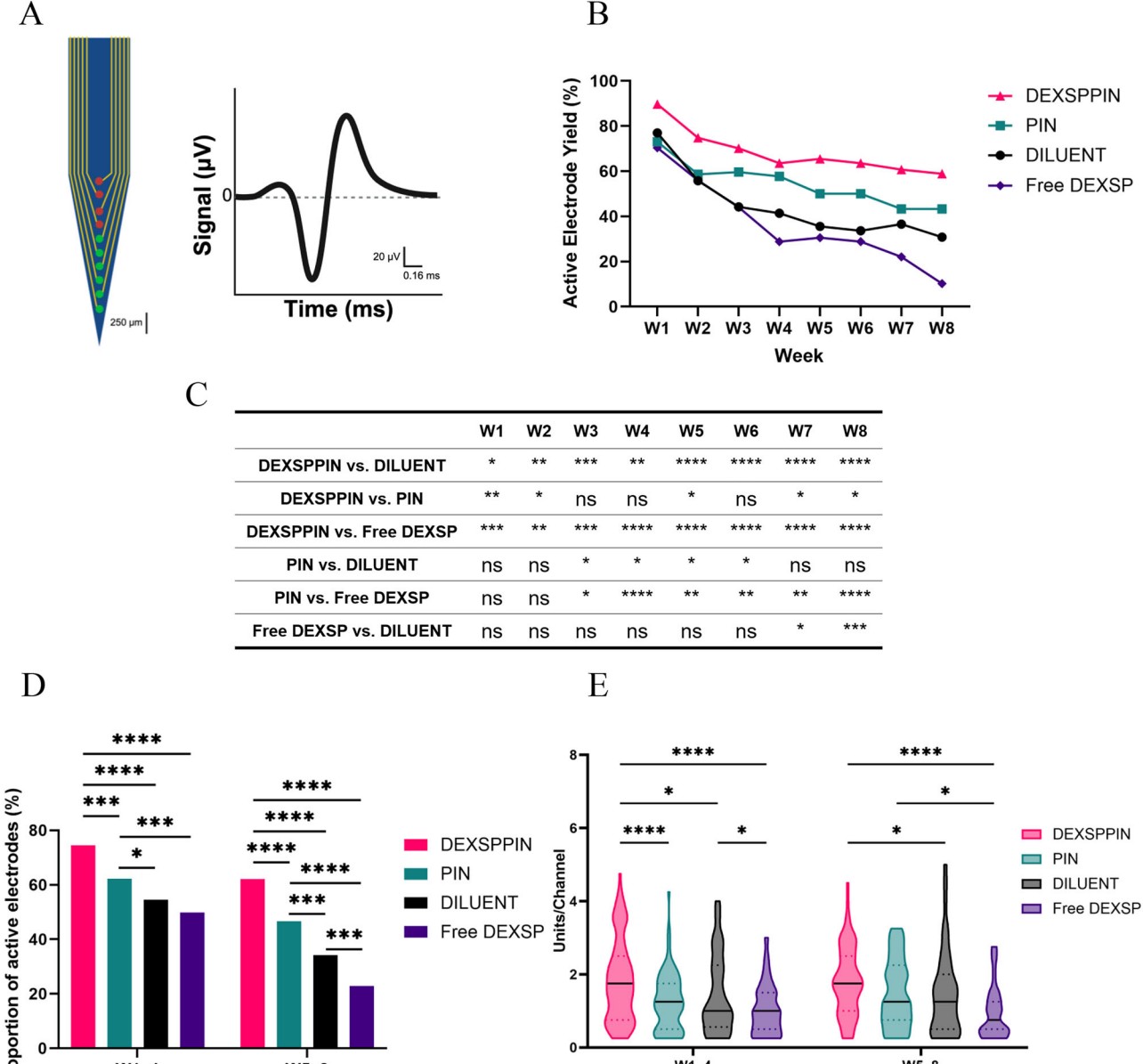

**Fig. 2 | In vivo single-unit electrophysiology to assess IME performance over 8 weeks. A** Schematic of the IME showing active (≥1 unit, green dot) and inactive (no unit, red dot) channels. Active electrode yield (AEY) is the number of active channels divided by the total number of usable channels per treatment per week[95]. **B** Weekly AEY for each treatment, with DEXSPPIN demonstrating significantly improved recording performance over PIN, DILUENT, and Free DEXSP. **C** Statistical comparisons (Two-way ANOVA (Mixed Model) with post hoc Tukey) of weekly AEY among groups, with significance levels indicated. **D** Proportion of active electrodes grouped by phase (W1–4 vs. W5–8), showing DEXSPPIN enhances recording stability in both early and late phases. **E** Active units per channel across treatments and phases. DEXSPPIN significantly outperforms other groups in W1–4; no significant difference from PIN in W5–8. The solid black line represents the median, and the colored dot indicates the interquartile range (25th to 75th percentile). Sample sizes: For **B**, determined by number of channels × number of animals: DEXSPPIN = 107, PIN = 104, DILUENT = 104, Free DEXSP = 118. For **D**, based on channels × weeks per phase × number of animals: DEXSPPIN = 428, PIN = 416, DILUENT = 416, Free DEXSP = 472. For **E**, based on total active channels per group per phase: DEXSPPIN W1–4 = 93, W5–8 = 51; PIN W1–4 = 77, W5–8 = 43; DILUENT W1–4 = 81, W5–8 = 38; Free DEXSP W1–4 = 86, W5–8 = 30. Malfunctioning channels from the outset were excluded. Significance: $p < 0.05$=*, $p < 0.01$=**, $p < 0.001$=***, $p < 0.0001$=****.

Interestingly, Vpp in the PIN group was significantly lower than that in the DILUENT group ($p < 0.05$), raising the possibility that the nanoparticle carrier alone might transiently suppress signal amplitude during the acute phase.

By the final four weeks (W5_8), Vpp values converged across all groups, with no significant differences detected, indicating a general decline in signal strength consistent with chronic implantation effects.

Noise levels were also evaluated as a determinant of signal fidelity. In W1_4, the DEXSPPIN group exhibited significantly higher noise compared to both the DILUENT ($p < 0.05$) and Free DEXSP ($p < 0.001$) groups (Fig. 3B). However, this elevation in noise did not persist into W5_8, suggesting that early treatment-related changes in the electrode microenvironment may have transiently elevated background activity or impedance.

Despite this, the signal-to-noise ratio (SNR) remained robust across all groups. Although SNR was significantly lower in the DEXSPPIN group compared to DILUENT ($p < 0.001$ in W1_4; $p < 0.0001$ in W5_8), the mean SNR in all cohorts exceeded 5 (Fig. 3C), a threshold generally considered indicative of high-quality single-unit recordings[50].

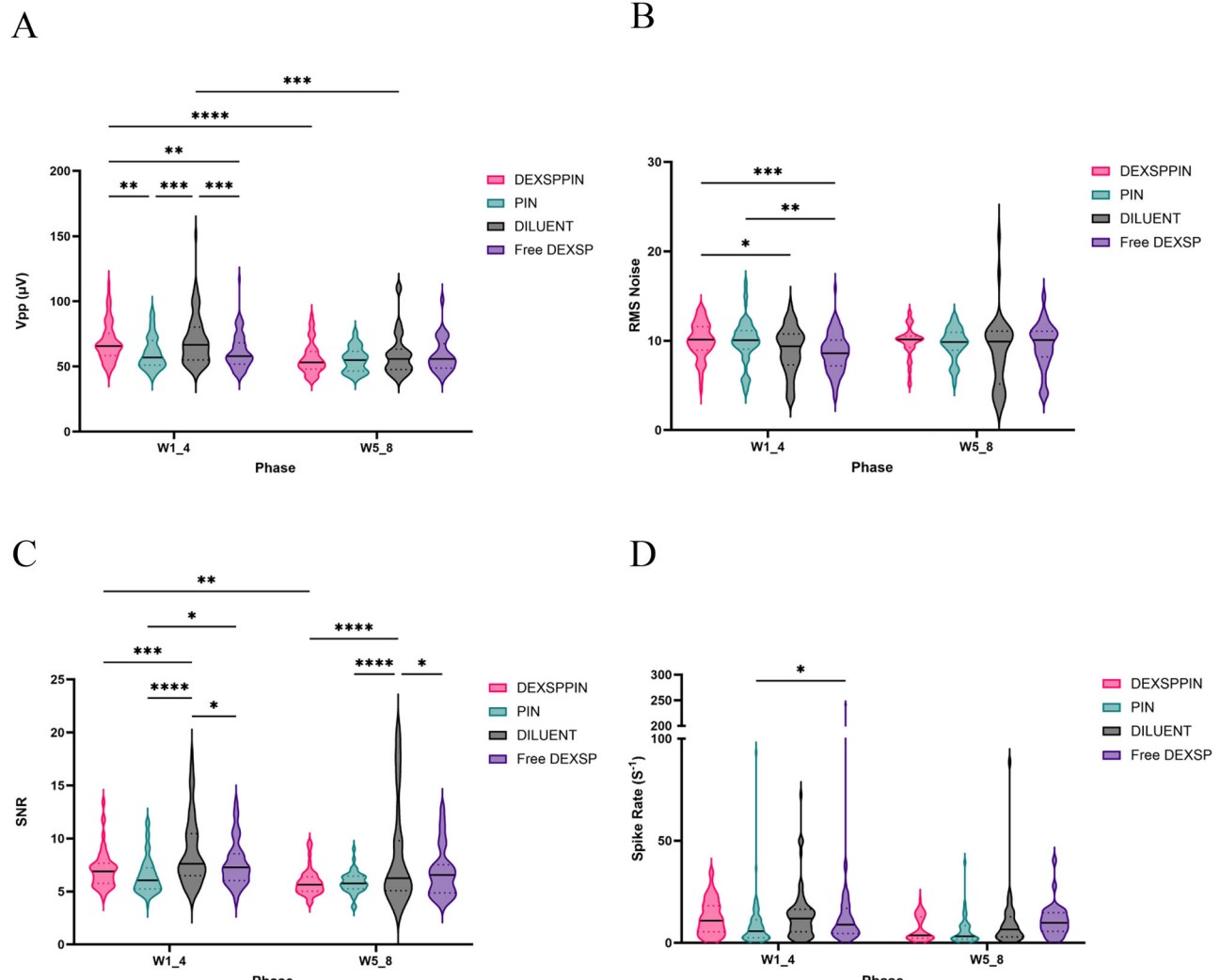

**Fig. 3 | Exported in vivo extracellular electrophysiology metrics. A** Peak-to-peak voltage of signals (Vpp)was significantly higher in the DEXSPPIN group compared to the PIN and Free DEXSP groups during the first four weeks of the study, no significant difference between the DEXSPPIN and DILUENT groups. **B** Significantly higher noise levels were observed in the DEXSPPIN group compared to the DILU-ENT and Free DEXSP groups during the first four weeks. No significant difference was observed between the DEXSPPIN and PIN groups. **C** Signal-to-noise ratio (SNR) was significantly lower in the DEXSPPIN group compared to the DILUENT group in both study phases. SNR was significantly lower in the PIN group compared to the

DILUENT and Free DEXSP groups. **D** Spike rate was significantly lower in the PIN group compared to the Free DEXSP group. No significant difference in spike rate was observed between DEXSPPIN and DILUENT groups in either phase. The sample size for **A–D** was determined by the total number of active channels across animals in each group over the entire phase: $n = 93$ for W1–4 DEXSPPIN, $n = 51$ for W5–8 DEXSPPIN, $n = 77$ for W1–4 PIN, $n = 43$ for W5–8 PIN, $n = 81$ for W1–4 DILUENT, $n = 38$ for W5–8 DILUENT, $n = 86$ for W1–4 Free DEXSP, and $n = 30$ for W5–8 Free DEXSP. Significance level is denoted as $p < 0.0001 = ****$, $p < 0.001 = ***$, $p < 0.01 = **$; $p < 0.05 = *$.

Lastly, we examined spike rates as an indirect measure of neuronal activity. The PIN group displayed significantly lower spike rates compared to the Free DEXSP group ($p < 0.05$), while no significant differences were observed between PIN and DILUENT. These findings suggest that continuous free drug exposure may modestly enhance neuronal firing rates. Notably, the DEXSPPIN group did not differ significantly from any other treatment group in spike rate, suggesting that targeted delivery preserved baseline firing patterns without overt hyper- or hypo-activation (Fig. 3D).

**Effects of DEXSPPIN on neuron density**

Insertion of IMEs causes acute neuronal loss and triggers a neuroinflammatory cascade that drives secondary neuron death, ultimately depleting the pool of viable neurons near the implant and impairing long-term recording performance. To assess the neuro-protective effects of DEXSPPIN, we quantified neuron density

around the electrode tract at the 8-week endpoint using NeuN immunostaining.

Representative images showed visibly greater neuronal pre-servation in the DEXSPPIN group compared to all controls (Fig. 4A). Quantitative analysis revealed that weekly systemic administration of DEXSPPIN significantly attenuated neuronal loss within 150 μm of the implant site, compared to the PIN, DILUENT, and Free DEXSP groups (Fig. 4B). Neuron counts were normalized to distal tissue regions (600–650 μm from the implant site) to control for inter-sample variability.

In the region closest to the implant (0–50 μm bin), the DEXSPPIN group retained 59% of baseline neuron density, representing a 186% increase over the DILUENT group, which retained only 21%. This pre-servation of local neuronal populations likely underlies the enhanced recording performance observed in DEXSPPIN-treated animals, as a higher density of viable neurons near the electrode interface increases the likelihood of capturing single-unit activity.

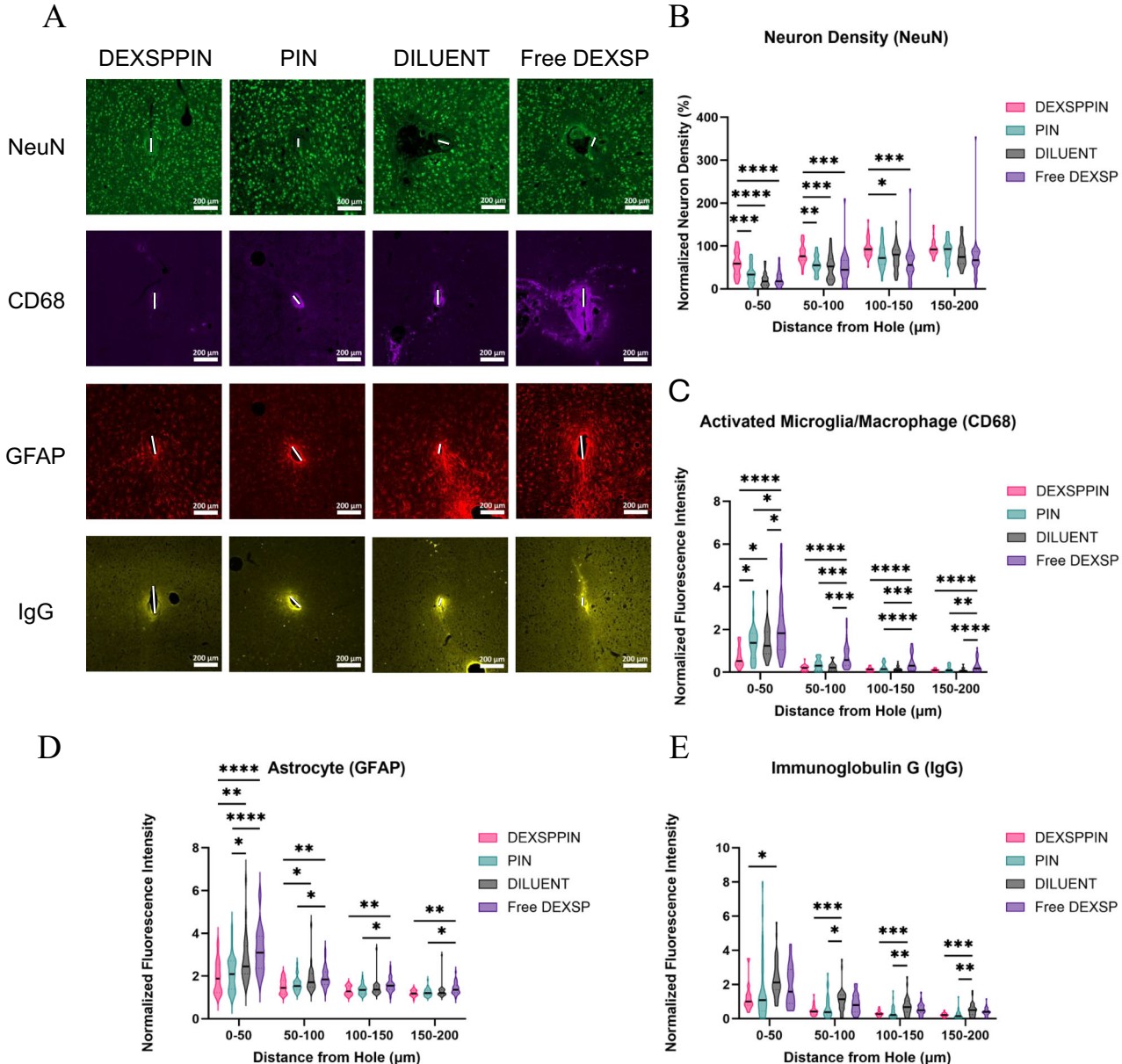

**Fig. 4 | Immunofluorescent intensities of NeuN, CD68, GFAP, and IgG near implant sites at 8-week endpoint. A** Representative false-colored images at 8 weeks. Scale bars = 200 μm. **B** Normalized neuron density vs. distance from the microelectrode interface. DEXSPPIN showed significantly higher neuron densities than all other groups up to 100 μm. **C–E** Normalized intensities of CD68, GFAP, and IgG as functions of distance. **C** CD68 was significantly lower in DEXSPPIN vs. all groups up to 50 μm. **D** GFAP was lower in DEXSPPIN vs. DILUENT and Free DEXSP up to 100 μm. **E** IgG was lower in DEXSPPIN vs. DILUENT up to 600 μm; PIN also reduced IgG vs. DILUENT from 50–600 μm. See Tables S3–S6 for full statistical comparisons. Violin plots show medians (black line) and interquartile ranges (colored dots). Neuron density and fluorescence intensities were normalized to values in the 600–650 μm bin per slice. Each biomarker analysis included ≥4 slides per treatment per animal, with each slide containing four randomly selected slices from varying depths. Statistics: One-way ANOVA with Tukey post hoc tests was used for each biomarker at each distance bin. Sample sizes (tissue slices per group): NeuN: DEXSPPIN = 29, PIN = 33, DILUENT = 40, Free DEXSP = 33; CD68: DEXSPPIN = 25, PIN = 29, DILUENT = 28, Free DEXSP = 39; GFAP: DEXSPPIN = 27, PIN = 32, DILUENT = 26, Free DEXSP = 30; IgG: DEXSPPIN = 24, PIN = 26, DILUENT = 30, Free DEXSP = 28. Tissue loss during staining contributed to slight variation in sample sizes. Significance: p < 0.05=*, p < 0.01=**, p < 0.001=***, p < 0.0001=****.

## Effects of DEXSPPIN on neuroinflammation and BBB permeability

Following IME insertion, both damage-associated molecular patterns (DAMPs), such as cellular debris from injured neurons, and pathogen-associated molecular patterns (PAMPs), such as blood-derived proteins, contribute to the activation and recruitment of microglia and macrophages near the implant site[51–54]. This sustained immune activation can lead to chronic neuroinflammation, glial encapsulation, and ultimately long-term failure of IME performance.

At the 8-week endpoint, DEXSPPIN-treated animals exhibited significantly reduced microglial activation, as assessed by CD68 staining, within 50 μm of the implant site compared to the PIN, DILUENT, and Free DEXSP groups (Fig. 4C). Notably, the Free DEXSP group exhibited the highest CD68 expression across all distances, indicating an exacerbated immune response despite systemic steroid delivery.

Astrocyte activation, quantified via GFAP staining, followed a similar trend. DEXSPPIN-treated animals showed significantly lower

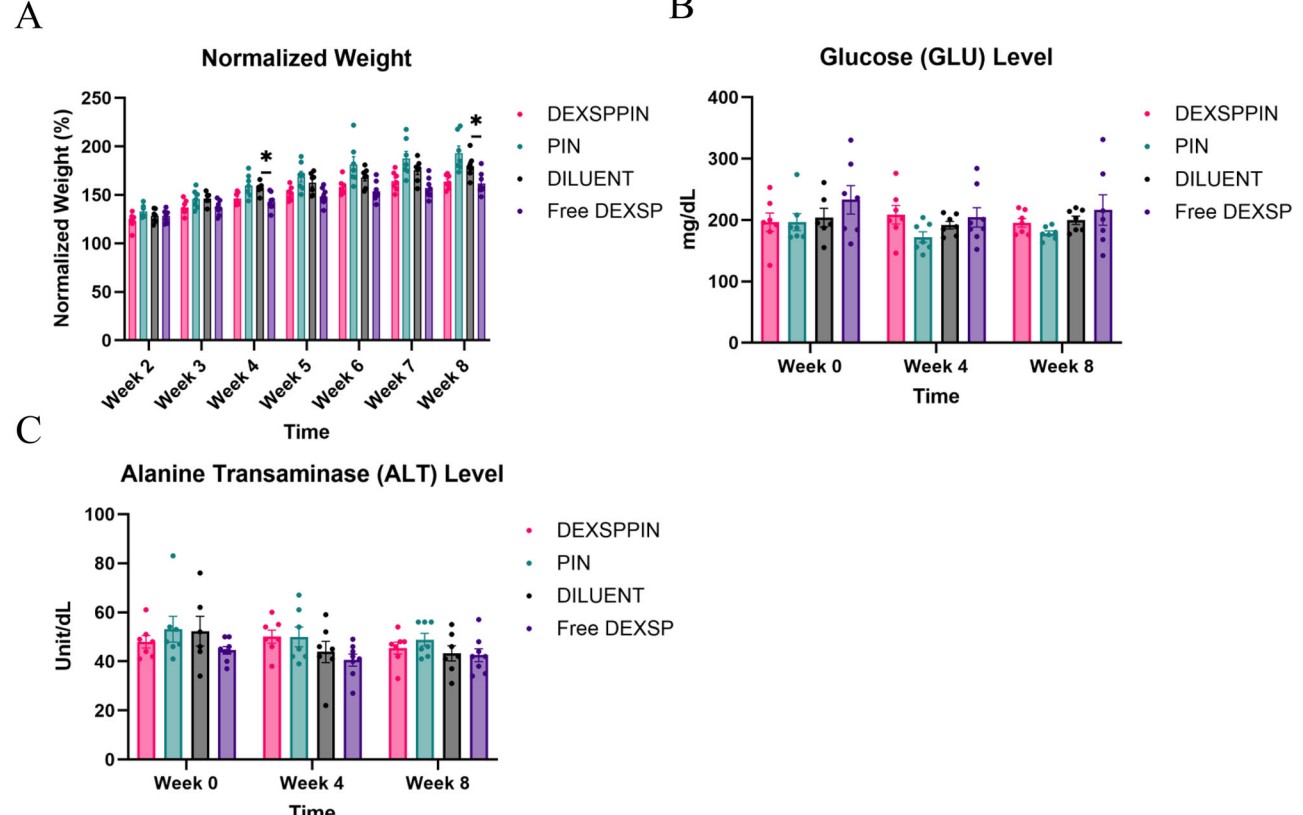

**Fig. 5 | Safety evaluations of weekly treatments over 8 weeks. A** Normalized body weights in the Free DEXSP group were significantly lower compared to the DILUENT group at week 4 and week 8, suggesting potential adverse effects on body weights from non-selective DEXSP treatment. **B** Glucose (GLU) levels showed no significant differences among all groups at week 0, week 4, and week 8. **C** Alanine transaminase (ALT) levels did not significantly vary among groups at week 0, week 4, and week 8, indicating no liver dysfunction. Kruskal-Wallis analyses with post hoc Dunn's pair-wise test were used to calculate statistical differences for all blood parameters and weight measurements. Sample sizes: DEXSPPIN: 7, PIN: 7, TH: 7, Free DEXSP: 8. Significance level is denoted as p < 0.05 = *. The standard error of the mean (SEM) was used to plot error bars for all plots.

GFAP expression up to 100 μm from the implant site relative to the DILUENT and Free DEXSP groups (Fig. 4D). PIN-treated animals also exhibited a modest but significant reduction in GFAP intensity up to 50 μm, suggesting a partial effect of the nanoparticle platform alone. The Free DEXSP group again showed the highest astrocyte activation, particularly within 50 μm of the implant site, aligning with elevated microglial responses in this group. Together, these findings indicate that DEXSPPIN treatment attenuates chronic neuroinflammation more effectively than free drug or vehicle controls.

To assess the potential of DEXSPPIN to promote blood-brain barrier (BBB) resealing, we stained for immunoglobulin G (IgG), a serum protein not normally present in the brain parenchyma and a widely used marker of BBB permeability. DEXSPPIN-treated animals exhibited significantly reduced IgG intensity up to 550 μm from the implant site compared to the DILUENT group (Fig. 4E; detailed significance thresholds provided in Table S6). A similar reduction in IgG was observed in the PIN group, with significantly lower levels extending from 50 μm to 600 μm from the implant site.

Although the DEXSPPIN and Free DEXSP groups did not differ significantly, a trend toward lower IgG levels was observed in the DEXSPPIN group. In contrast, Free DEXSP-treated animals displayed IgG intensities comparable to the DILUENT group at all distances, suggesting limited impact on BBB repair. These results support the hypothesis that targeted DEXSP delivery via the PIN platform enhances BBB resealing and reduces post-insertion microvascular permeability more effectively than free drug administration alone.

## Side effects of DEXSPPIN on metabolic homeostasis and organ functions

Chronic systemic administration of dexamethasone is associated with several adverse effects, including hyperglycemia, weight loss, and reduced bone mineral density in rodent models[45]. To evaluate the safety of repeated systemic DEXSPPIN administration, we monitored markers of metabolic function (body weight, glucose), liver function (alanine transaminase, ALT), and kidney function (creatinine, CREA) over the course of the study. Body weights were recorded weekly, and glucose, ALT, and CREA levels were assessed at baseline (week 0), midpoint (week 4), and study endpoint (week 8). DEXSPPIN-treated animals maintained stable body weights and glucose levels comparable to the DILUENT control group throughout the study (Fig. 5A, B), suggesting no significant metabolic disruption. In contrast, animals treated with Free DEXSP showed significant weight loss at both week 4 and week 8 compared to DILUENT controls, indicating a possible adverse metabolic effect from repeated free drug administration. Liver function, assessed via serum ALT levels, showed no significant elevation in the DEXSPPIN group relative to DILUENT-treated animals (Fig. 5C), supporting the hepatic safety of the formulation. Kidney function, measured by serum CREA, remained within normal limits for all groups. Most CREA values were below the assay's detection threshold, indicating no detectable nephrotoxicity over the treatment period (Table S7). Collectively, these findings indicate that weekly systemic administration of DEXSPPIN over 8 weeks does not induce significant metabolic, hepatic, or renal toxicity, while modest effects were observed with Free DEXSP.

## Discussion

Intracortical microelectrodes are essential tools for neuroscience research and brain-machine interfaces aimed at restoring motor function after limb loss or spinal cord injury[55–59]. However, their long-term utility is limited by a rapid decline in recording performance, often within weeks of implantation. This decline is primarily driven by insertion-induced neuroinflammation and blood-brain barrier disruption, which allow blood-derived proteins and immune triggers to enter the brain parenchyma[60–65].

Insertion trauma inevitably causes bleeding and tissue damage, triggering microglial and astrocytic activation, proinflammatory cytokine release (e.g., TNF-α, IL-1β), and cytotoxic species such as reactive oxygen and nitrogen species[63,65]. These factors contribute to secondary neuronal damage and glial encapsulation, leading to chronic device failure.

Systemic and local dexamethasone delivery has been explored to attenuate these effects, but both approaches have limitations. Systemic administration suffers from poor bioavailability and off-target effects, while local delivery typically requires device modification and lacks sustained control. Few studies have addressed the opportunity to promote hemostasis and reseal the BBB to prevent inflammatory triggers from entering the brain.

In this study, we evaluated a strategy combining injury-targeted delivery and hemostatic activity via platelet-inspired nanoparticles (PINs) loaded with dexamethasone sodium phosphate (DEXSP). We hypothesized that systemic DEXSPPIN would enhance drug localization at the implant site, promote BBB resealing, and synergistically attenuate neuroinflammation to improve IME recording performance.

Our results confirmed this hypothesis. Weekly administration of DEXSPPIN over eight weeks significantly preserved active electrode yield and increased the number of recordable single units compared to controls. The proportion of active electrodes was 36% higher in weeks 1–4 and 82% higher in weeks 5–8 in DEXSPPIN-treated animals versus DILUENT controls. This improved recording performance was accompanied by significantly lower IgG levels near the implant, suggesting reduced BBB permeability and restricted influx of blood-derived proteins.

Histological analyses further supported the proposed mechanism. DEXSPPIN-treated animals exhibited significantly reduced microglial (CD68) and astrocytic (GFAP) activation near the implant site compared to all other groups. Neuron density within $50 \mu m$ of the IME was nearly threefold higher in the DEXSPPIN group than in the DILUENT group, likely contributing directly to the improved electrophysiological outcomes.

The PIN-only group showed moderate improvements in recording performance and reduced IgG infiltration, suggesting that the nanoparticles alone may aid BBB resealing. However, as IgG is an indirect marker and can also reflect inflammation-induced permeability, we cannot conclusively determine whether PINs directly stabilize the BBB. Future work should explore this using endothelial markers and real-time imaging.

Systemic Free DEXSP treatment worsened recording performance relative to DILUENT, consistent with prior studies showing that non-targeted DEX can have paradoxical effects[38,41]. For instance, Chen et al. demonstrated that dexamethasone improved outcomes in septic mice but impaired wound healing in healthy animals[66], which may explain the exacerbated inflammation in our otherwise healthy rats. This underscores the importance of spatially targeted anti-inflammatory therapy.

Mechanistically, we propose that DEXSP released from PINs binds to glucocorticoid receptors on activated microglia, suppressing NF-κB signaling through IκBα upregulation[67–69]. This may disrupt the self-sustaining inflammatory loop at the implant site, reducing cytokine production, limiting astrocyte recruitment, and preventing secondary neuron death. Although we were unable to directly quantify proinflammatory cytokines due to technical limitations with fixed tissue,

future studies could apply spatial transcriptomics or NanoString analysis to explore local gene expression profiles[70].

We observed no adverse effects from weekly DEXSPPIN administration. Glucose, ALT, and creatinine levels remained within normal limits, and body weights were stable throughout the study. In contrast, the Free DEXSP group experienced transient weight loss, highlighting the safety advantage of targeted delivery. Previous biodistribution studies showed PIN accumulation in liver and kidney, raising concerns about chronic organ toxicity. However, no evidence of liver or kidney dysfunction was found in our 8-week study.

Despite these encouraging results, long-term safety remains a key consideration for the clinical translation of DEXSPPIN. Chronic glucocorticoid use is associated with well-documented side effects, including osteoporosis, hyperglycemia, and weight loss[45]. Moreover, PINs are designed to promote hemostasis[71], which may carry prothrombotic risk during extended use. While Hickman et al. reported no thrombosis in major organs at a 1-hour timepoint post-PIN administration[49], further studies are needed to assess clotting risk with chronic dosing.

Translational safety is further complicated by species-specific differences in nanomedicine responses (Szebeni 2020). This variability underscores the importance of testing in diverse models. Encouragingly, the base nanoparticle formulation—SynthoPlate® (Haima Therapeutics)—has already demonstrated safety and efficacy in multiple preclinical species, including mouse, rat, rabbit, and pig, with titrated effective dosing established in several studies[46,72–74]. In our own 8-week study, no severe adverse reactions or gross behavioral changes were observed in rats, but longer-term studies in additional species will be essential for fully assessing biocompatibility and systemic effects.

Another key challenge for translation lies in the method of delivery. Although intra-arterial injection was initially chosen to avoid complications such as phlebitis from repeated tail vein access, this route is not practical in clinical settings due to the increased risk of vascular injury and embolic events, particularly when administering nanoparticle-based formulations[75]. Future studies should investigate the durability of therapeutic effects with shorter or lower dosing regimens, explore safer and more clinically feasible delivery routes, and validate both efficacy and safety in large animal models to support translational readiness.

This study demonstrates that targeted delivery of DEXSP via platelet-inspired nanoparticles significantly improves the long-term recording performance of intracortical microelectrodes. By combining injury-targeted drug delivery with hemostatic activity, DEXSPPIN mitigates neuroinflammation, preserves neuron density, and reduces BBB permeability near the implant site. In contrast, non-targeted Free DEXSP worsened inflammation and signal quality, emphasizing the need for localized therapeutic approaches. While the PIN group alone demonstrated some benefit in recording performance over the DILUENT control, it was modest compared to the DEX-loaded DEXSPPIN group. Future studies should focus on refining dosing strategies, characterizing molecular mechanisms, and evaluating long-term safety to advance the clinical translation of this promising neuroprotective strategy.

## Methods

### Ethics

Every experiment involving animals was carried out following a protocol approved by an ethical commission. The studies described herein were reviewed and approved by the Institutional Animal Care and Use Committee (IACUC) at the Louis Stokes Cleveland Department of Veteran Affairs Medical Center and were performed in accordance with the ARRIVE guidelines.

### Intracortical microelectrode validation and sterilization

The quality of 16-channel, single-shank intracortical microelectrode (A1x16-3mm-100-177-Z16, NeuroNexus, Ann Arbor, MI) was validated

using electrochemical impedance spectroscopy (EIS) assays. A Gamry Interface 1010E Potentiostat (Gamry Instruments, Warminster, PA, USA)was employed to perform EIS on each electrode site of the IME, using the electrode site as the working electrode, Ag | AgCl electrode as the reference and a platinum wire as the counter electrode. EIS measurements were conducted in 1 × PBS electrolyte (pH 7.5) and all wire connections placed inside a Faraday cage. Impedances at each electrode site were tested across a frequency range of 1 HZ to $10^6$ Hz (12 points per decade) in a 50 mV rms AC voltage. The 1 kHz impedance was specifically measured to validate the quality of IME prior to in vivo single-unit electrophysiology recording. IMEs with normal 1 kHz impedance values (between 100 kΩ and 1 MΩ) across all 16 electrode sites were deemed suitable for IME implantation surgery. Validated IMEs and 2 mm × 123 μm × 15 μm non-functional Michigan-style single-shank silicon microelectrodes (Qualia Labs, Dallas, TX) were sterilized by Ethylene Oxide (EtO) prior to surgeries.

### Intracortical microelectrode implantation surgery

Twenty-nine male rats (Sprague Dawley, CD, Charles River Labs, Wilmington, MA) with indwelling vascular catheters weighing 250-280 grams, were included in this study. Surgeries to implant vascular catheters were performed by Charles River Labs. In this study, polyurethane catheters were inserted into the carotid artery and advanced to the aortic arch for weekly treatments[46]. All rats were housed for recovery for up to 1 week prior to further procedures.

Surgical procedures for implanting functional IMEs in the rat cortex were adapted from previously published methods[22,46,76,77]. All rats were initially anesthetized with 3.5% isoflurane and maintained at a surgical level of anesthesia with 1.5-3% isoflurane throughout the procedure. After ensuring a surgical anesthetic depth was achieved, eye ointment was applied to protect the corneas, and the nails were clipped to reduce the risk of premature removal of postoperative sutures. The surgical site was shaved, and the animals received subcutaneous injections of cefazolin (16 mg/kg) and meloxicam (1 mg/kg) to prevent infection and manage pain, respectively. Additionally, 0.2 mL of 0.25% Marcaine was administered locally to the incision site for localized pain relief. Following these preparations, the rat was secured in a stereotaxic frame using blunt ear bars and an incisor bar. Vital parameters, including SpO2, heart rate, and body temperature, were continuously monitored throughout the surgery. Regular toe pinches and observations of paw color were performed to confirm the maintenance of an adequate anesthetic plane. The incision area was sterilized with alternating applications of Betadine and isopropyl alcohol.

Under sterile conditions, a midline scalp incision (approximately 25 mm) was made to expose the skull for craniotomy procedures. Hydrogen peroxide was applied to the exposed skull to clean the area by removing blood and connective tissue, creating a dry surface. Butyl cyanoacrylate (VetBond, 3 M, Saint Paul, MN) was used to control bleeding, maintain skull dryness, and enhance the adherence of the headcap. Craniotomies for stainless steel bone screws (Stoelting Co., Wood Dale, IL) for ground wires (−1.5 mm anterior/posterior, −1.5 mm lateral to bregma) and reference wires (−5.5 mm anterior/posterior, −1.5 mm lateral to the bregma) were performed using a dental drill with 1.35 mm diameter drill bit. Craniotomies for functional IMEs (+ 2 mm anterior/posterior, +3 mm lateral to the bregma) and non-functional IMEs (+ 2 mm anterior/posterior and −3 mm lateral to the bregma) were performed with 1.75 mm diameter drill bit (see schematic in Fig. 6A). Once the dura mater was exposed from drilling, a dura pick was used to reflect the dura. Ground and reference wires on functional IMEs were wrapped and secured on screws before implantation of IMEs. Once the positions were secured, the functional IME was inserted to the motor cortex with 2 mm depth using the stereotaxic frame. Non-functional IMEs were manually implanted at approximately 2 mm depth for improving the sample size (N) for the histology study. Excess

bone debris and blood from drilling and insertion were then cleaned by applying saline, followed by the application of Kwik-cast to seal the empty space in the craniotomies with implants. After allowing the Kwik-cast to cure for approximately 15 minutes, Teets Cold Cure dental cement was applied around all craniotomies to build a headcap for stabilizing all implants and fuse the base to the skull. After the dental cement was cured, 5-0 monofilament polypropylene sutures were used to suture the incision site. Cefazolin and meloxicam were subcutaneously administered to the animal for two-days as post-operative treatment for infection and pain.

### Dexamethasone sodium phosphate loaded nanoparticle manufacture and characterization

DEXSPPIN were manufactured with the 'thin-film rehydration and extrusion' method adapted from previously established protocols[47,49,78,79]. Briefly, DSPC, cholesterol, DSPE-mPEG(1k), DSPE-PEG(2k)-FMP, DSPE-PEG(2k)-VBP, DSPE-PEG(2k)-CBP and DSPE-Cy5 were homogeneously mixed in 1:1 chloroform:methanol organic solvent at 46.50, 45.00, 6.50, 0.50, 0.25, 0.25 and 1.00 mole percentages, respectively. After the thin lipid film was formed using a rotary evaporator, the lipid thin film was rehydrated with 5% w/v trehalose in 4.05 mg/mL HEPES solution (DILUENT) containing 4 mg dexamethasone sodium phosphate. Ten freeze/thaw cycles were applied to the rehydrated lipid solution with liquid nitrogen and a 75 °C water bath to maximize drug encapsulation, followed by extrusion with 200 nm then 100 nm polycarbonate membranes using a pneumatic extruder (Evonik, Burnaby, Canada) to form small unilamellar vesicles. Nanoparticle characterizations, including hydrodynamic size distribution, zeta potential, and polydispersity index (PDI) of DEXSPPIN, were conducted using Data Litesizer (Anton Paar, Ashland, VA). Zeta potential measurements were performed using an Omega cuvette (Mat. No. 225288, Anton Paar, Ashland, VA) and analyzed under the Smoluchowski approximation with a Henry factor of 1.5. Measurements were conducted using water as the solvent, with the refractive index set to 1.33 and relative permittivity at 78.3. Endotoxin levels in DEXSPPIN were evaluated using the Endosafe Limulus Amebocyte Lysate (LAL) kit (#3672151, Charles River Laboratories, Charleston, SC).

### Dexamethasone sodium phosphate loaded nanoparticles purification and encapsulation efficiency characterization

Unencapsulated DEXSP was separated from DEXSPPIN using a centrifugation method. Freshly manufactured DEXSPPIN was transferred to a 10 kDa molecular weight cutoff (MWCO) Amicon filter (Millipore-Sigma, Burlington, MA, UFC801024) in a 15 mL Falcon tube, followed by an initial centrifugation at 3200 x g for 20 minutes. To wash the DEXSPPIN trapped on the filters, 0.5 mL of DILUENT solution was added, and a second centrifugation was performed to remove any remaining unencapsulated DEXSP. The mass of the separated unencapsulated DEXSP was quantified via reverse-phase high-performance liquid chromatography (RP-HPLC, Shimadzu). The analysis utilized a C18 reverse-phase column (Shimadzu, Columbia, MD, #220-91199-13) and a UV-vis detector (SPD40V, Shimadzu, Columbia, MD).

For RP-HPLC analysis, unknown encapsulated DEXSP samples and DEXSP standards were prepared using desoximetasone (DES) as an internal standard to account for variations in injection and HPLC system performance. The mobile phase consisted of an isocratic mixture of 32:68 v/v acetonitrile/10 mM phosphate buffer, with an injection volume of 30 μL and a flow rate of 1 mL/min. UV absorbance was measured at 240 nm, with retention times of 3.5 minutes for DEXSP and 22 minutes for DES. Standard calibration curves of the DEXSP-DES peak area ratios and DEXSP-DES concentration ratios were plotted using OriginLab software (OriginLab Corporation, Northampton, MA) to determine unknown DEXSP concentrations. Encapsulation

A

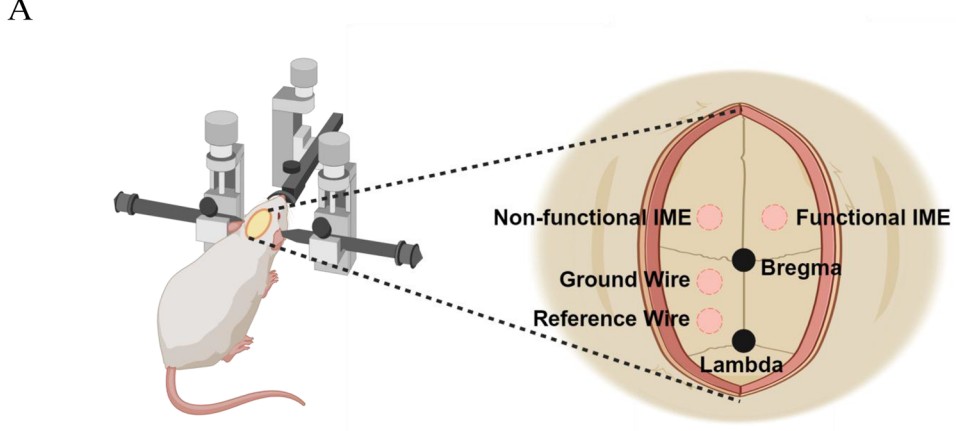

B

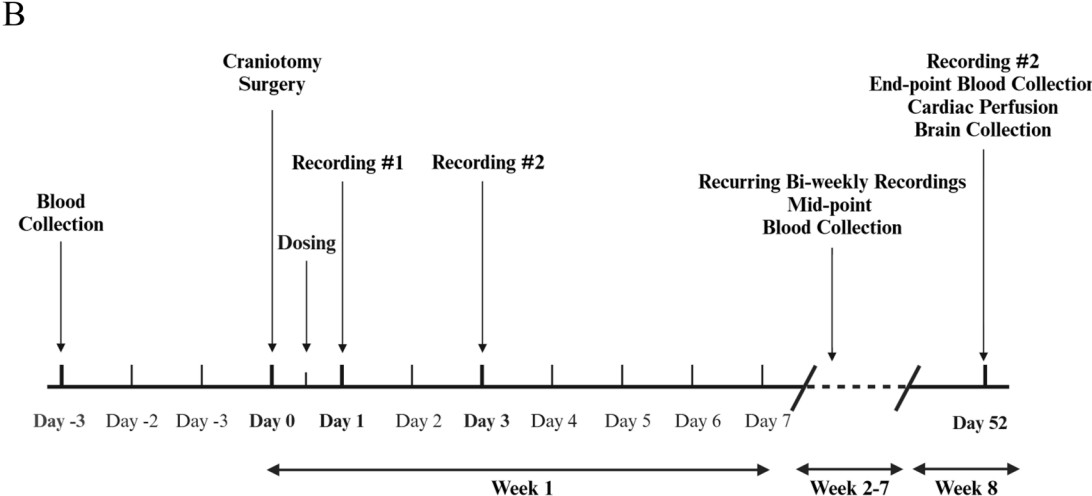

**Fig. 6 | Schematic of craniotomy, implantation, and study timeline.**
**A** Craniotomies were performed at the labeled sites, showing different types of implantations: non-functional IME, functional IME and bone screws for ground and reference wires. Non-functional IMEs were inserted to increase sample size (N) for the histology study[96]. **B** Timeline of the 8-week study, including blood collection, craniotomy surgery, weekly treatment dosing, bi-weekly neural recording, and 8-week endpoint procedures for cardiac perfusion and brain collection[97].

efficiencies were calculated using Eq. (1).

$$Encapsulation\ Efficiency(\%)$$
$$= \frac{[Total\ drug\ (mg) - Unencapsulated\ drug\ (mg)]}{Total\ drug\ (mg)} \times 100\% \quad (1)$$

**Dexamethasone sodium phosphate in vitro release**
A dialysis membrane (DM) method was employed to characterize the in vitro release profile of DEXSP from DEXSPPIN[80]. Briefly, 0.5 mL of purified DEXSPPIN solution was transferred into a dialysis device (Slide-A-Lyzer™ MINI Dialysis Devices, Thermo Fisher Scientific, Waltham, MA) with a 10 kDa MWCO. The device was submerged in 14 mL of 1× PBS within a 15 mL Falcon tube and incubated at 37 °C in a shaker set to 70 rpm. At specified time points, 1 mL aliquots were collected from the dialysis medium containing released DEXSP for quantification using RP-HPLC analysis. After each sampling, an equal volume of fresh 1× PBS was added to maintain the dialysis conditions.

**Dexamethasone sodium phosphate loaded nanoparticles administration**
Twenty-nine catheterized rats were randomly assigned to groups to receive weekly administrations of one of the following treatments: dexamethasone sodium phosphate-loaded platelet-inspired nanoparticles (DEXSPPIN, n = 7, 2 mg/kg nanoparticle dosage and 0.3 mg/kg equivalent drug dosage), platelet-inspired nanoparticles (PIN, n = 7, 2 mg/kg nanoparticle dosage), free dexamethasone sodium phosphate (DEXSP, n = 8, 0.3 mg/kg drug dosage), or 5% w/v trehalose in 4.05 mg/mL HEPES solution (DILUENT, n = 7) as diluent control. The summary of treatments was shown in Table 1. The administration timeline is shown in Fig. 6B. The infusion method via indwelling vascular catheters was performed according to a previously established protocol[46]. Surgeries to implant vascular catheters were performed by Charles River Labs. Briefly, sterile syringes containing the treatments were connected to external tether tubing with PinPort injectors to access the carotid artery catheters. The treatments were infused towards the aortic arch at a rate of 0.1 mL/min using a syringe pump. Following the infusion, the catheters were maintained by flushing with 500 units/mL heparin to prevent clogging.

**Table 1 | Summary of in vivo treatment groups**

| Treatment | Drug dosage (mg/kg) | Nanoparticle dosage (mg/kg) | Frequency | N of animals |
|---|---|---|---|---|
| DEXSPPIN | 0.3 | 2 | Once per week | 7 |
| PIN | N/A | 2 | Once per week | 7 |
| DEXSP | 0.3 | N/A | Once per week | 8 |
| DILUENT | N/A | N/A | Once per week | 7 |

### Electrophysiological recordings

Neural electrophysiology recordings were conducted twice per week for 8 weeks using functional intracortical microelectrodes (IMEs) implanted in rats adapted from previous reported methods[16,17,81–84]. The rats were initially anesthetized with 3.5% isoflurane and maintained under surgical anesthesia with 1.5-3% isoflurane during the cleaning of the exposed IME connectors to remove blood and tissue debris using isopropanol. Following this, the rats were transferred into a custom-made acrylic box within a Faraday cage while still under anesthesia. A 16-channel ZIF-Clip Headstage (Tucker-Davis Technologies Inc., Alachua, FL) was attached to both sides of the IME connectors, which were connected to the 32-channel Motorized Commutator System (#ACO32, Tucker-Davis Technologies Inc.). This setup allowed the rats to move freely and minimized motion artifacts. A 16-Channel Medusa Pre-Amp (RA16PA, Tucker-Davis Technologies Inc.) was connected to the commutator to amplify and digitize the analog signals, which were then transmitted to a Bioamp Processor (RZ5, Tucker-Davis Technologies Inc.) via fiber optic connectors for further signal processing. The processed signals were displayed and recorded using Synapse software (Tucker-Davis Technologies Inc.) on a computer. Neural recordings were conducted for 10 minutes after the rats awoke, at a sampling rate of 24.414 kHz with a built-in 300-3000 Hz bandpass filter to capture single-unit activity[85]. After the 10-minute recording session, the rats were disconnected from the headstage and returned to their housing cages.

### Electrophysiological analysis

Recorded digital neural signals were analyzed using Plexon Offline Sorter (Plexon Inc., Dallas, TX) to identify single units[17,77]. The data were first referenced to the common median across all 16 channels of the IMEs to eliminate potential artifacts arising from the rats' movement and common-mode signals. Spikes were detected using −4σ threshold from the mean of the peak heights. After spike detection, any spikes with amplitudes exceeding ±500 μV were considered abnormal neuronal signals and were excluded from the analysis. Additionally, spikes detected across more than 14 channels were classified as motion artifacts caused by the rats' grooming or movement and were removed. Following artifact removal, automatic sorting was performed using the K-Means Scan method to cluster the remaining spikes into individual single units. True positive signal units were manually verified following previously established protocols[84]. The active electrode yield in each treatment group for each week was calculated by dividing the total number of active channels (defined as those with at least one active unit) by the total number of channels (16 channels per IMEs device × number of animals for each treatment in the specific week, excluding channels that never have any active units). The proportion of active channels for each treatment group in each time phase (the first four weeks, W1-4, and the last four weeks, W5-8) was calculated similarly, but by summing the total active and total channels across all weeks in the respective time phase. On a weekly basis, the total number of channels (N) was as follows: 107 for DEXSPPIN, 104 for PIN, 104 for DILUENT, and 118 for Free DEXSP. For the entire phase, the total number of channels (N) was 428 for DEXSPPIN, 416 for PIN, 416 for DILUENT, and 472 for Free DEXSP.

Peak-to-peak voltage (Vpp) was defined as the absolute voltage range from the positive to negative peak of a clustered and verified single unit. Noise was calculated as the root mean square (RMS) of the signal after spike exclusion. Signal-to-noise ratio (SNR) was determined by dividing the Vpp by the noise for each verified single unit. Spike rate was defined as the inverse of the median interspike interval for each verified single unit. Calculated recording parameters were obtained using custom-developed MATLAB software (Mathworks, Natick, MA). Units with a Vpp less than 40 μV were considered non-putative and excluded from further analysis[83]. The calculated recording parameters were grouped into two intervals: the first four weeks of the study (W1-4) and the last four weeks (W5-8). Outliers in SNR values across all treatments and phases were identified using the Robust Regression and Outlier Removal (ROUT) method in GraphPad Prism, with a False Discovery Rate (FDR) set to 5% (Q = 5%). The SNR data were fit using a nonlinear regression model that reduces the influence of outliers. Residuals, calculated as the differences between observed and predicted values, were then analyzed to detect significant deviations using the FDR approach, allowing accurate outlier identification while controlling multiple comparisons[86]. Units with verified SNR outliers were excluded from further analysis. Vpp, noise, SNR, and spike rate values for each unit were averaged at the individual channel level for each IME within different treatment groups and time phases, resulting in a single averaged value of Vpp, noise, SNR, and spike rate for each active channel. The total number of units for each active channel was counted and divided by the number of weeks in the specific time phase to calculate the units per active channel. This yielded a single average number of units per active channel for each IME across different treatment groups and time phases, which was used for graph plotting and statistical analysis. For Vpp, noise, SNR, spike rate, and units per active channel data, the sample size (N) represents the total number of active channels for different treatment groups at different phases: 93 for W1-4 DEXSPPIN, 51 for W5-8 DEXSPPIN, 77 for W1-4 PIN, 43 for W5-8 PIN, 81 for W1-4 DILUENT, 38 for W5-8 DILUENT, 86 for W1-4 Free DEXSP, 30 for W5-8 Free DEXSP.

### Animal perfusion and tissue processing

At the 8-week endpoint, rats were administered an intraperitoneal overdose of ketamine (160 mg/kg) and xylazine (20 mg/kg). Cardiac perfusion and dissection procedures were adapted from a previously established method[87]. In brief, rats were perfused with 600-700 mL of 1× PBS at a flow rate of 50-70 mL/min. Once the liquid flowing from the right atrium was clear, an additional 300-400 mL of 10% neutral buffered formalin solution was perfused. The brains were then dissected and stored in 10% neutral buffered formalin at 4 °C for 24 hours, followed by cryoprotection using a 30% w/v sucrose solution containing 0.01% sodium azide. A 10% stepwise increase in sucrose concentration was performed every 24 hours, with two exposures at 30%. The cryoprotected brains were subsequently embedded in Optimal Cutting Temperature (OCT) compound (Scigen Tissue-Plus™ O.C.T. Compound, Fisher Scientific, Hampton, NH, USA) and frozen on dry ice, then stored at −80 °C. Brain sections were cut transversely at 20 μm thickness using a cryostat (CM1950, Leica, Deer Park, IL).

### Immunohistochemistry, imaging and analysis

Immunohistochemistry (IHC) staining was performed on brain slices to visualize and assess neuron loss, microglial activation, astrocyte activation, and BBB permeability (antibodies listed in Table S1). IHC procedures were adapted from previously reported methods[24,37,46,88,89]. Standard positive and negative controls were run to validate each of the antibodies used in our study[37]. Brain slices on slides were incubated for 15 minutes in a humidified chamber at room temperature, followed by washing with 1× PBS to remove the O.C.T. compound. The tissues were rehydrated and permeabilized with 1× PBS containing 0.1% Triton for 15 minutes. To prevent non-specific antibody binding, tissues were

blocked in a 4% goat serum solution. Primary antibodies were applied and incubated overnight. On the following day, tissues were washed multiple times with 1× PBS containing 0.1% Triton to remove any remaining primary antibodies. Secondary antibodies were then applied for 2 hours. Afterward, a series of washing steps were performed to eliminate any residual secondary antibodies and Triton, followed by the application of DAPI Fluoromount-G mounting medium and coverslips. The slides were dried overnight and stored at 4 °C for imaging.

Slides with stained brain slices were imaged using the Automatic Slide Scanner Axioscan Z7 (Zeiss Inc., Oberkochen, Germany) with the 20× objective. One slide from each type of staining was used to determine the optimal exposure time. The optimal exposure times were consistent over the entire study. The resulting images were subset around the implantation region and exported as 16-bit TIFF images (10,000 × 10,000 pixels by pixels) using ZEN 3.6 Blue Edition software for fluorescent intensity analysis. A custom Python program, based on SECOND[90], was employed to define implant holes and mask image artifacts. Concentric rings were generated by the program, starting from the outer edge of the implant hole and extending across the image in 50 μm intervals, to measure the average fluorescence intensities in each bin as a function of distance from the hole. Intensities in masked regions due to artifacts were excluded from the measurements[91]. As reported in previous methods[46], average fluorescence intensities for each bin were normalized to the background intensity in the 600-650 μm bin. This bin was selected as an internal reference region because it is deemed sufficiently far away from the microelectrode implantation site and, based on our histological assessment, does not exhibit signs of implant-induced neuroinflammation or neurodegeneration[76]. Therefore, the 600–650 μm range was considered representative of healthy background tissue for normalization purposes. Since CD68-positive microglia and immunoglobulin G (IgG) are not expected in healthy brain parenchyma, their normalization factors were set to 0, whereas the factor for astrocytes (GFAP), which are normally present, was set to 1.

To count NeuN stained neurons around the implant hole, Cellpose algorithm was used to segment the neurons on NeuN stained slices[92]. As reported in a previous established workflow[93], a specialized cyto3-based model was further trained with at least one stained slide from each treatment group using default parameters. All NeuN images were segmented in batches using the trained model. A custom Python script calculated neuron density from the raw NeuN segmentation masks. Similar to the intensity analysis, neuron density (# of neuron per mm$^2$) at each 50 μm bin was calculated by dividing neuron counts by the total area of each bin, starting from the implant site, and normalized to the density in the 600-650 μm bin.

For each IME implant hole, at least three different brain slice depths were analyzed for both the SECOND fluorescent intensity analysis and the Cellpose neuron density analysis. For each treatment group, brain tissues from a minimum of four animals were included in both analyses.

### Blood collection and analysis
At weeks 0 (prior to treatment), 4, and 8 (following weekly treatments), rats were anesthetized with 3.5% isoflurane and maintained under a surgical plane of anesthesia (1.5-3% isoflurane). As described in a previously published method[46]. 300 μL of blood was then collected from the tail vein using a capillary blood collection tube (Multivette® 600 Lithium heparin LH, Sarstedt, Germany). After plasma was separated from whole blood by centrifugation, glucose (GLU), creatinine (CREA), and alanine transaminase (ALT) levels were measured in the plasma.

### Statistics
Excel (Microsoft Corporation, Redmon, WA), R Studio 2023.12.0 + 369 (RStudio, PBC, Boston, MA) and GraphPad Prism (Dotmatics, Boston, MA) were used for data analysis, rearrangement and statistical

measurements. Custom R scripts were used to rearrange in vivo extracellular single unit recording raw data, including the number of units and active channel yield across different animals, treatments, and weeks. A two-sided proportions z-test was performed using an R-based script to calculate significant statistical differences in active electrode yield between groups on a weekly basis, as well as the proportion of active electrode between and within groups at different study phases. Statistical differences in exported recording parameters (Vpp, noise, SNR, spike rate) were calculated using a mixed-effects model analysis with post hoc Tukey pair-wise comparisons in GraphPad. A one-way ANOVA with post hoc Tukey pair-wise test (α = 0.05) in GraphPad was used to compare normalized neuron density (NeuN) and normalized fluorescence intensity for astrocyte (GFAP), activated microglia/macrophage (CD68), and Immunoglobulin G (IgG) among treatment groups within each distance interval. The Kruskal-Wallis test with post hoc Dunn's pair-wise test in GraphPad was conducted to calculate statistical differences in blood parameters and weight measurements. Statistical significance was represented as * = p < 0.05, ** = p < 0.01, *** = p < 0.001 and **** = p < 0.0001 for all statistical analyses in this study. The standard error of the mean (SEM) was used to plot error bars for all bar plots.

### Reporting summary
Further information on research design is available in the Nature Portfolio Reporting Summary linked to this article.

### Data availability
The drug release, electrophysiology, immunohistochemistry, and blood testing data generated in this study have been deposited in a github repository https://github.com/Shofflab/SPPINDEX. Source data are provided with this paper.

### Code availability
Code is available at the following repository under CC-BY-NC license: https://github.com/Shofflab/SPPINDEX.

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

## Acknowledgements
This research was funded by the U.S. Department of Veterans Affairs (I01RX003420 and GRANT12635707)—A.J.S., J.R.C. This work was also supported by the National Institute of Health R01 (HL121212—A.S.G.), and the National Institute of Biomedical Imaging and Bioengineering (T32EB004314—D.M.M.). The views, interpretations, conclusions, and recommendations expressed are solely those of the authors and do not necessarily reflect those of the U.S. Department of Veterans Affairs, the National Institutes of Health, or the United States Government. BioRender.com was used to create schematics for Figs. 1A, 2A and 6 included in this manuscript.

## Author contributions
L.L.: writing—original draft, writing—review & editing, visualization, investigation, methodology, formal analysis, data curation, validation, conceptualization. A.H.: investigation, data curation. D.M.M.: investigation, conceptualization. J.Z.: data curation, software. A.C.: investigation. D.V.L.: conceptualization. B.T.: writing—review & editing, resources. E.Q.: resources. D.N.: writing—review & editing. G.F.H.: methodology. C.L.P.: writing—review & editing, resources, conceptualization, funding acquisition. M.A.B.: writing—review & editing, resources, conceptualization, funding acquisition. A.S.G.: writing—review & editing, supervision, resources, conceptualization, funding acquisition. J.R.C.: writing—review & editing, supervision, conceptualization, funding acquisition. A.J.S.: writing—review & editing, conceptualization, project administration, funding acquisition, supervision, software.

## Competing interests
B.T., E.Q., C.P., M.B. are employees of Haima Therapeutics LLC. A.S.G. is a co-founder and chief scientific adviser for Haima Therapeutics, a pre-clinical stage biopharma company developing platelet-inspired therapies, including SynthoPlate™, to mitigate active bleeding and bleeding risks after traumatic injury, surgery, thrombocytopenia, and rare bleeding disorders. The remaining authors declare no competing interests.
