## [Transparent Peer Review file · Nature Communications]

Dexamethasone-loaded platelet-inspired nanoparticles improve intracortical microelectrode recording performance

Corresponding Author: Dr Andrew Shoffstall

Version 0:

Reviewer comments:

Reviewer #1

(Remarks to the Author)

Li et al. show results supporting an innovative method (i.e., systemic administration of SPPINDEX) to improve the stability and reliability of chronic microelectrode recordings by stimulating repair of the blood-brain barrier. The authors present a characterization of the DEX-loaded nanoparticles that mimic platelets to promote blood clotting. The data support improved neural recording quality 8 weeks after implant and increased neuron density while reducing activated microglia, astrocytes and BBB permeability.

Major comments:

- 1) The authors added rigor to the study by examining side effects of SPPINDEX in rats up to the longest time point in the study (8 weeks post-implant). Please comment on any potential side effects and/or other safety concerns of SPPINDEX systemic administration in more chronic administration beyond 8 weeks. Also, would any difference in side effects of SPPINDEX systemic administration be expected humans vs. rats? For example, could blood clots formed outside of the injury site increase risk of heart attack or stroke? Have the authors explored other modes of delivery more local to the insertion site to circumvent concerns due to systemic delivery?
- 2) Include more details on the following methods to ensure other groups can replicate the results. In "electrophysiological analysis": Provide more details on the ROUT method. In "IHC, Imaging and Analysis": SECOND not defined
- 3) The fact that free DEX worsened recording performance was an interesting result that highlights the importance of the BBB repair aspect of SPPINDEX. Were there any specific indicators from the brain tissue of the free DEX group that suggest delayed wound healing (consistent with the findings of Chen et al. as mentioned in the Discussion section)?

Minor comments:

- BBB is repeatedly defined
- Fix reference 46
- Figure 1: Not clear that neural recordings continue during weeks 2-7 from figure alone, though methods spell this out.
- Figure 3A: Include scale bar for electrode and for spike amplitude and time

Reviewer #2

(Remarks to the Author)

The manuscript presents a new approach to mitigating neuroinflammatory responses associated with IME implantation using dexamethasone-loaded platelet-inspired nanoparticles (SPPINDEX). The study is well-structured and provides compelling evidence that SPPINDEX treatment enhances IME recording performance over eight weeks in a rat model. The integration of neuroinflammation mitigation with a targeted drug delivery system is a noteworthy advancement. However, there are areas where clarity, methodological rigor, and interpretation of results could be improved.

Figures

- Several figures are not referred to in the main text, making it unclear how they support the study's claims. Every figure should be explicitly mentioned and discussed within the appropriate section.
- The order of figures is inconsistent, with some figures missing in the text and others appearing out of sequence. This should be carefully reviewed and corrected to improve readability.
- Figure 3D appears to be identical to Figure 3F-I, with only a different version of the graphs (different grouping but based on the same dataset). If this is the case, the authors should clarify why both are included and whether one can be removed or merged for simplicity.
- In Figure 5, the normalization strategy for neuron density and fluorescence intensity should be explicitly stated in the figure legend. Normalization to background intensity or an internal reference should be clarified. The choice of the 600–650 μm bin for normalization should be justified.

Statistical Analysis

- The manuscript does not consistently indicate the number of animals (n) used in each experimental group. The sample size for each experiment should be clearly stated in figure legends.
- The meaning of each statistical annotation should be explicitly described in the figure legends. The corresponding p-values should be provided in each figure legend to ensure clarity for readers.

Results

- The study presents neuron density as an indicator of SPPINDEX's therapeutic effect (Figure 5). However, increased neuron density alone may not be a definitive marker to claim a 'therapeutic' effect of SPPINDEX. I suggest revising the wording to simply state it as the 'effect of SPPINDEX' rather than implying a therapeutic impact without additional functional validation.
- SPPINDEX was administered weekly for 8 weeks. However, it's unclear whether continuous dosing is necessary for sustained improvements in IME recording performance or if a gradual reduction in dosing could still provide long-term benefits while minimizing potential side effects.
- The role of hemostatic properties in BBB sealing should be expanded upon. Is there direct evidence that PIN promotes endothelial repair?

Discussion

- The initial portion of the Discussion repeats content that was already covered in the Introduction. This redundancy reduces the impact of the discussion and makes the manuscript unnecessarily lengthy.
- The discussion of dexamethasone side effects is appreciated, but additional data on inflammatory markers in peripheral organs (e.g., liver, spleen) would strengthen claims of systemic safety.

I recommend major revisions before acceptance for publication.

Reviewer #3

(Remarks to the Author)

See attached comment document.

Version 1:

Reviewer comments:

Reviewer #1

(Remarks to the Author)

Thank you for your email. The authors have satisfactorily responded to all of my concerns and I believe that the paper will make an important contribution to the field.

(Remarks on code availability)

Reviewer #2

(Remarks to the Author)

Minor Revisions

1. Referring to Figure 6B in the middle of the Results section (while discussing Figure 2) is confusing. Since it's a Methods figure, consider rephrasing as "see timeline in Methods" instead of citing it directly.
2. In the Methods section, the schematic is incorrectly referred to as Figure 5A, whereas it now appears as Figure 6A.
3. Supplementary Tables (S3–S7): The group labels in these supplementary tables still reflect old naming conventions. These should be updated to match the revised treatment group names.

(Remarks on code availability)

Reviewer #3

(Remarks to the Author)

The authors thoroughly addressed my comments (and the other reviewers'). One small comment: the doi for citation 90 is not complete. Additionally, the github does not appear to have all the data available.

(Remarks on code availability)

The code available is an .exe file. To access the .m file for running in Matlab, it was easier to access via reference 90 in the supplemental information. Reference 90 is essentially the readme file.

We thank the reviewers for their excellent feedback and the opportunity to revise our manuscript. We have made substantial formatting changes throughout and streamlined our introduction and discussion sections according to the comments provided. Clarifications and additional details were added throughout. Some notable structural changes are noted below:

- 1) We moved the Methods section to its proper location after the Discussion section.
- 2) Consequently, we substantially revised our Results section to provide additional context and rationale.
- 3) We moved the original Figure 1 to follow the Discussion section, and it is now labeled as Figure 6. As a result, the original Figures 2–6 have been renumbered as Figures 1–5. All figure references in this response document reflect the new order of figures in the manuscript.
- 4) We adjusted the following acronyms for clarity:
 - We changed “SPPINDEX” to “DEXSPPIN” (dexamethasone sodium phosphate loaded platelet-inspired nanoparticles) to more accurately reflect the order of the words in the acronym.
 - We changed “TH” (trehalose solution) to “DILUENT” to improve clarity in results interpretation.

Point-by-point response

Reviewer #1 (Remarks to the Author):

Li et al. show results supporting an innovative method (i.e., systemic administration of SPPINDEX) to improve the stability and reliability of chronic microelectrode recordings by stimulating repair of the blood-brain barrier. The authors present a characterization of the DEX-loaded nanoparticles that mimic platelets to promote blood clotting. The data support improved neural recording quality 8 weeks after implant and increased neuron density while reducing activated microglia, astrocytes and BBB permeability.

Major comments:

1) The authors added rigor to the study by examining side effects of SPPINDEX in rats up to the longest time point in the study (8 weeks post-implant). Please comment on any potential side effects and/or other safety concerns of SPPINDEX systemic administration in more chronic administration beyond 8 weeks. Also, would any difference in side effects of SPPINDEX systemic administration be expected humans vs. rats? For example, could blood clots formed outside of the injury site increase risk of heart attack or stroke? Have the authors explored other modes of delivery more local to the insertion site to circumvent concerns due to systemic delivery?

Safety beyond 8 weeks: To answer your question directly, we have not yet tested particle administration in vivo beyond the 8-week time point in this study (or others) and therefore can't claim to know what the longer term effects may be. There are multiple potential risks including those related to chronic steroid use (e.g., decreased bone density, compromised immune system), chronic nanomedicine use (e.g., antibody formation, or as the reviewer suggests perhaps thrombosis or clotting), and simply chronic intravenous administration (injection site infection, pain, abscesses).

To answer your question a little differently, we believe that chronic, indefinite, systemic administration of our DEXSPPIN nanoparticle is likely not a viable solution for multiple reasons: cost, convenience, and the potential (yet untested) risk factors typically associated with its chronic administration. However, **our data suggests that transient 2-8 week courses of treatment may potentially be sufficient to provide initial neuroprotection that yields sustained benefits in recording performance.** Figure 2B in our manuscript depicts a recording performance decline that is most severe in the first four weeks after implantation and begins to plateau thereafter. This suggests that early neuroprotection may result in sustained benefits.

The question around the longer-term durability of the effects from an early course of treatment is the subject of a pending grant proposal. Therefore, it falls within “intended future studies” rather than within the scope of our present study. Thus, in this manuscript, we believe that 8-weeks is a relevant time point for our present state of our development and for evaluation of the therapy. We agree with the reviewer that there is need to assess the long-term safety of this nanomedicine prior to clinical testing.

We have added relevant portions of this answer to the discussion section:

“Despite these encouraging results, long-term safety remains a key consideration for the clinical translation of DEXSPPIN. Chronic glucocorticoid use is associated with well-documented side effects, including osteoporosis, hyperglycemia, and weight loss [45]. Moreover, PINs are designed to promote hemostasis [90], which may carry pro-thrombotic risk during extended use. While Hickman et al. reported no thrombosis in major organs at a 1-hour timepoint post-PIN administration [49], further studies are needed to assess clotting risk with chronic dosing.”

Species differences: There are known safety differences between species, especially as they relate to nanomedicines (Szebeni 2020). This is something that we will encounter as we continue to translate this therapy. However, it is very encouraging that the base formulation of these particles SynthoPlate® (Haima Therapeutics) has already been successfully tested in mouse, rat, rabbit and pig animal models, with titrated effective doses (Dyer 2018, Li 2024, Srinivasan 2024, Hickman 2018). We have added this point to the discussion.

“Translational safety is further complicated by species-specific differences in nanomedicine responses (Szebeni 2020). This variability underscores the importance of testing in diverse models. Encouragingly, the base nanoparticle formulation—SynthoPlate® (Haima Therapeutics)—has already demonstrated safety and efficacy in multiple preclinical species, including mouse, rat, rabbit, and pig, with titrated effective dosing established in several studies (Dyer 2018, Li 2024, Srinivasan 2024, Hickman 2018). In our own 8-week study, no severe adverse reactions or gross behavioral changes were observed in rats, but longer-term studies in additional species will be essential for fully assessing biocompatibility and systemic effects.”

Routes of administration and thrombotic risks: We intend to switch to intravenous injections for future work. The present studies used an intra-arterial route due to the availability of rats with indwelling intra-arterial lines to avoid complications from repeated tail-vein injections. However, intravenous administration is likely to be more translatable. We have also performed preliminary studies using nose-to-brain delivery route. However, the nanoparticles developed here are not optimized for that route, and did not yield a measurable concentration of particles around the implant site. There are numerous other catheter-based local brain drug delivery solutions, however, those require complex device modifications. A motivation of our study was to find a targeted drug delivery solution that could avoid these requirements.

Notably, Hickman *et al.* reported no thrombotic risk in the lung, liver, kidney, or spleen following platelet-inspired nanoparticle administration at a 1-hour time point. However, multi-organ thrombotic risk assessment has not yet been performed following chronic administration. While no severe reactions resulting in gross behavioral changes in the rats were observed during the course of our study, additional safety testing is warranted in the future. This point is included in the discussion,

“Another key challenge for translation lies in the method of delivery. Although intra-arterial injection was initially chosen to avoid complications such as phlebitis from repeated tail vein access, this route is not practical in clinical settings due to the increased risk of vascular injury and embolic events, particularly when administering nanoparticle-based formulations. [91] Future studies should investigate the durability of therapeutic effects with shorter or lower dosing regimens, explore safer and more clinically feasible delivery routes, and validate both efficacy and safety in large animal models to support translational readiness.”

Dyer MR, et al. Intravenous administration of synthetic platelets (SynthoPlate) in a mouse liver injury model of uncontrolled hemorrhage improves hemostasis. *J Trauma Acute Care Surg.* 2018. doi: 10.1097/TA.0000000000001893. PMID: 29538234; PMCID: PMC5970031.

Hickman, D.A., *et al.* Intravenous synthetic platelet (SynthoPlate) nanoconstructs reduce bleeding and improve 'golden hour' survival in a porcine model of traumatic arterial hemorrhage. *Sci Rep* (2018). <https://doi.org/10.1038/s41598-018-21384-z>

Li L, Menendez-Lustri DM, Hartzler A, Pogharian A, Zaorski B, Chen A, Palen J, Traylor B, Quill E, Pawlowski CL, Bruckman MA, Gupta AS, Capadona JR, Shoffstall AJ. Systemically administered platelet-inspired nanoparticles to reduce inflammation surrounding intracortical microelectrodes. *Biomaterials*. 2025 Jun;317:123082. doi: 10.1016/j.biomaterials.2025.123082. Epub 2025 Jan 2. PMID: 39787896.

Srinivasan AJ, Secunda ZA, Mota-Alvidrez RI, Luc NF, Disharoon D, Traylor B, Pawlowski CL, Brown JB, Bruckman MA, Gupta AS, Neal MD. Platelet-inspired synthetic nanoparticles improve hemostasis and hemodynamics in a rabbit model of abdominal hemorrhage. *J Trauma Acute Care Surg*. 2024 Jan 1;96(1):101-108. doi: 10.1097/TA.0000000000003938. Epub 2023 Dec 7. PMID: 38057963; PMCID: PMC10746291.

Szebeni J, Bawa R. Human Clinical Relevance of the Porcine Model of Pseudoallergic Infusion Reactions. *Biomedicines*. 2020 Apr 8;8(4):82. doi: 10.3390/biomedicines8040082. PMID: 32276476; PMCID: PMC7235862.

2) Include more details on the following methods to ensure other groups can replicate the results. In “electrophysiological analysis”: Provide more details on the ROUT method. In “IHC, Imaging and Analysis”: SECOND not defined

The details of ROUT method in GraphPad were added to method section: “Electrophysiological Analysis”:

Outliers in SNR values across all treatments and phases were identified using the Robust Regression and Outlier Removal (ROUT) method in GraphPad Prism, with a False Discovery Rate (FDR) set to 5% (Q = 5%). The SNR data were fit using a nonlinear regression model that reduces the influence of outliers. Residuals, calculated as the differences between observed and predicted values, were then analyzed to detect significant deviations using the FDR approach, allowing accurate outlier identification while controlling multiple comparisons (Motulsky 2006).

For histological image quantification, we used a custom Python-based software tool adapted from our previously developed MATLAB-based platform, referred to as **SECOND**. The core quantification principles are outlined in our earlier publication: “*A graphical user interface to assess the neuroinflammatory response to intracortical microelectrodes*” (Lindner 2019).

In brief, the Python version of SECOND was employed to define implant holes and mask image artifacts. Concentric rings were generated by the program, starting from the outer edge of the implant hole and extending across the image in 50 µm intervals, to measure the average fluorescence intensities in each bin as a function of distance from the hole. Intensities in masked regions due to artifacts were excluded from the measurements. **The paragraph above was added to the methods section.**

To promote reproducibility and transparency, the Python-based version of the SECOND software used in this study has been uploaded to Zenodo for open access. [<https://doi.org/10.5281/zenodo.15603085>]

S.C. Lindner, et al, A graphical user interface to assess the neuroinflammatory response to intracortical microelectrodes HHS Public Access, *J Neurosci Methods* 317 (2019) 141–148. <https://doi.org/10.1016/j.jneumeth>.

H.J. Motulsky, R.E. Brown, Detecting outliers when fitting data with nonlinear regression - A new method based on robust nonlinear regression and the false discovery rate, *BMC Bioinformatics* 7 (2006). <https://doi.org/10.1186/1471-2105-7-123>.

3) The fact that free DEX worsened recording performance was an interesting result that highlights the importance of the BBB repair aspect of SPPINDEX. Were there any specific indicators from the brain tissue of the free DEX group that suggest delayed wound healing (consistent with the findings of Chen et al. as mentioned in the Discussion section)?

In our study, animals treated with free DEX showed significantly elevated levels of reactive astrocytes and activated microglia around the implant site compared to other treatment groups (Figure 4C and D). This suggested immune cell response likely contributed to impaired neural recording performance, as a thicker encapsulation layer around the implant can increase impedance. The abundance of activated immune cells at the 8-week time point suggests a persistent inflammatory environment, indicating that weekly free DEX treatment

at 0.3 mg/kg may have disrupted the normal wound healing process or even exerted a proinflammatory effect. While the exact mechanism underlying this response cannot be fully determined from this study, our ongoing omics-based analysis suggests that repeated dosing of free DEX may induce a rebound activation of inflammatory pathways. These data are currently being prepared for publication.

Minor comments:

-BBB is repeatedly defined – Revised accordingly.

-Fix reference 46 – Fixed.

-Figure 1: Not clear that neural recordings continue during weeks 2-7 from figure alone, though methods spell this out. – We have adjusted the figure #1 to figure #6 because of moving method after discussion. We have included a revised Figure 6 which adds an additional bar to denote this.

-Figure 3A: Include scale bar for electrode and for spike amplitude and time. Fixed – now Figure 2A.

Reviewer #2 (Remarks to the Author):

The manuscript presents a new approach to mitigating neuroinflammatory responses associated with IME implantation using dexamethasone-loaded platelet-inspired nanoparticles (SPPINDEX). The study is well-structured and provides compelling evidence that SPPINDEX treatment enhances IME recording performance over eight weeks in a rat model. The integration of neuroinflammation mitigation with a targeted drug delivery system is a noteworthy advancement. However, there are areas where clarity, methodological rigor, and interpretation of results could be improved.

Figures

- Several figures are not referred to in the main text, making it unclear how they support the study's claims. Every figure should be explicitly mentioned and discussed within the appropriate section. The order of figures is inconsistent, with some figures missing in the text and others appearing out of sequence. This should be carefully reviewed and corrected to improve readability.

Thank you for pointing out this oversight. We have reviewed all figures and ensured that each contributes meaningfully to the manuscript. We removed several panels to the supplementary, and re-organized others to improve the flow of the manuscript. All remaining figures and panels have sequential in-text references, accordingly. The original Figure 1 has been moved to follow the Discussion section and is now labeled as Figure 6. As a result, the original Figures 2–6 have been renumbered as Figures 1–5. Specifically: Figure 1, we adjusted the order of the panel presentation; Figure 2, we removed panels F-I and placed them in supplementary (also discussed in the next comment); Figure 5, we changed the order of the panel presentation.

- Figure 3D appears to be identical to Figure 3F-I, with only a different version of the graphs (different grouping but based on the same dataset). If this is the case, the authors should clarify why both are included and whether one can be removed or merged for simplicity.

We agree with the reviewer's comment. Figure 2F-I were originally included to highlight the statistical significance of changes in the proportion of active electrodes across different phases within each treatment group. These were presented as separate panels due to limited space in Figure 2D for indicating statistical comparisons. To improve clarity and streamline Figure 2, we have moved Figure 2F-I to the supplementary materials (Figure S1A – S1D), as they are derived from the same dataset as Figure 2D but serve a different interpretative purpose.

- In Figure 4, the normalization strategy for neuron density and fluorescence intensity should be explicitly stated in the figure legend. Normalization to background intensity or an internal reference should be clarified. The choice of the 600–650 μm bin for normalization should be justified.

The following text was added to the Figure 4 legend, “Neuron density and fluorescence intensity values were normalized to the values within the 600–650 μm bin for each individual brain slice”

The following text was also added to the methods section, “This bin was selected as an internal reference region because it is deemed sufficiently far away from the microelectrode implantation site and, based on our histological assessment, does not exhibit signs of implant-induced neuroinflammation or neurodegeneration (Shoffstall 2018). Therefore, the 600–650 μm range was considered representative of healthy background tissue for normalization purposes.”

A.J. Shoffstall, M. Ecker, V. Danda, A. Joshi-Imre, A. Stiller, M. Yu, J.E. Paiz, E. Mancuso, H.W. Bedell, W.E. Voit, J.J. Pancrazio, J.R. Capadona, Characterization of the neuroinflammatory response to thiol-ene shape memory polymer coated intracortical microelectrodes, *Micromachines* (Basel) 9 (2018). <https://doi.org/10.3390/mi9100486>.

Statistical Analysis

- The manuscript does not consistently indicate the number of animals (n) used in each experimental group. The sample size for each experiment should be clearly stated in figure legends.

For Figure 1:

The sample size for release study was added in caption 1 (D): (n = 3).

For Figure 2:

The sample size for B was determined by the total number of channels multiplied by the number of animals: n = 107 for DEXSPPIN, n = 104 for PIN, n = 104 for DILUENT, and n = 118 for Free DEXSP. The sample size for D was determined by the total number of channels multiplied by the number of weeks in each phase and the number of animals in each group: n = 428 for DEXSPPIN, n = 416 for PIN, n = 416 for DILUENT, and n = 472 for Free DEXSP. Channels that were malfunctioning from the beginning were excluded from the analysis. The sample size for D was determined by the total number of channels multiplied by the number of weeks in each phase and the number of animals in each group: n = 428 for SPPINDEX, n = 416 for PIN, n = 416 for DILUENT, and n = 472 for Free DEXSP. Channels that were malfunctioning from the beginning were excluded from the analysis.

For Figure 3:

The sample size for A-D was determined by the total number of active channels across animals in each group over the entire phase: n = 93 for W1–4 SPPINDEX, n = 51 for W5–8 SPPINDEX, n = 77 for W1–4 PIN, n = 43 for W5–8 PIN, n = 81 for W1–4 DILUENT, n = 38 for W5–8 DILUENT, n = 86 for W1–4 Free DEXSP, and n = 30 for W5–8 Free DEXSP.

For Figure 4:

For the IHC staining, a minimum of two slides per treatment per animal were included for each biomarker, with each slide containing four slices randomly selected from various tissue depths. The total number of slices analyzed varied by biomarker due to occasional tissue loss during the staining process. Sample sizes for IHC analysis were as follows: NeuN – DEXSPPIN: 29, Free PIN: 33, TH: 40, Free DEXSP: 33; CD68 – DEXSPPIN: 25, Free PIN: 29, TH: 28, Free DEXSP: 39; GFAP – DEXSPPIN: 27, Free PIN: 32, TH: 26, Free DEXSP: 30; IgG – DEXSPPIN: 24, Free PIN: 26, TH: 30, Free DEXSP: 28.

For Figure 5:

Sample sizes: DEXSPPIN: 7, PIN: 7, TH: 7, Free DEXSP: 8.

- The meaning of each statistical annotation should be explicitly described in the figure legends. The corresponding p-values should be provided in each figure legend to ensure clarity for readers.

Corresponding p-values for different significance levels were added in all figure captions: “Significance level is denoted as $p < 0.0001 = ****$, $p < 0.001 = ***$, $p < 0.01 = **$; $p < 0.05 = *$ ”.

Results

- The study presents neuron density as an indicator of SPPINDEX's therapeutic effect (Figure 4). However, increased neuron density alone may not be a definitive marker to claim a 'therapeutic' effect of SPPINDEX. I suggest revising the wording to simply state it as the 'effect of SPPINDEX' rather than implying a therapeutic impact without additional functional validation.

We agree with the reviewer that the increased neuron density characterized by NeuN biomarker staining is not a definitive sign to indicate the therapeutic effect of SPPINDEX. The title of the result section “Therapeutic Effects of SPPINDEX on Neuron Health and Density” was changed to “Effects of SPPINDEX on Neuron Density”. All terms like “Neuron Health” and “Therapeutic Effects of SPPINDEX” were changed to “Neuron Density” and “Effects of SPPINDEX” in the manuscript.

- SPPINDEX was administered weekly for 8 weeks. However, it's unclear whether continuous dosing is necessary for sustained improvements in IME recording performance or if a gradual reduction in dosing could still provide long-term benefits while minimizing potential side effects.

This is a great suggestion and is a part of ongoing work in our lab. As alluded to in our response to reviewer #1, we have a pending grant submission to evaluate the durability of these observed neuroprotective effects after a transient (2-8 week) course of treatment. We do not yet have sufficient evidence to answer this question, but have included the following statement in the discussion section:

“Future studies should investigate the durability of therapeutic effects with shorter or lower dosing regimens, explore safer and more clinically feasible delivery routes, and validate both efficacy and safety in large animal models to support translational readiness.”

- The role of hemostatic properties in BBB sealing should be expanded upon. Is there direct evidence that PIN promotes endothelial repair?

The present study evaluates the performance of each of the therapies with regards to in vivo recording quality and end-point histological tissue response. Unfortunately, the present study was not specifically designed to tease apart whether the primary mechanism of PIN-Only's effects was either *direct* stabilization of the BBB re-sealing or indirect stabilization of the BBB by otherwise reducing the nearby neuroinflammation, which has known effects on the permeability of the BBB.

Future studies may consider incorporating other BBB-focused assessments such as labeled dextrans or Evans Blue dye in order to evaluate whether the acute treatment of the nanoparticles has such effects on high molecular weight compound extravasation, or whether these develop over time. If the BBB permeability reduces acutely with PIN-only treatment, it may suggest a more direct mechanism for stabilizing the BBB. However, if the effects only develop over several weeks, it gives more credence to a more nuanced or indirect mechanism.

Furthermore, we have now collected separate unpublished data that is supportive of potential our PIN's direct influence on the BBB. To further elucidate potential mechanisms of action, we performed bulk transcriptomics of the implant site tissue to reveal major changes in the activated molecular pathways (Menendez, in preparation). Specifically, we examined genes involved in endothelial and brain vasculature repair (Fn1, PEAR1, and Pink1), genes involved in tight and cadherin junction (Cdh5, Cldn5 and Tjp1), and genes involved in pericyte recruitment

(Pdgfrb and Tgfb1). The strongest mechanistic indicator that we observed that supported non-drug-loaded PIN alone may be helping to stabilize the BBB was determined from the observation that Pdgfrb expression surrounding the IME implant was increased compared to naïve controls, whereas in a diluent-treated control it was not (Log2FoldChange = 0.33, p<0.05).

PDGFR-β is a receptor that binds to platelet-derived growth factor (PDGF) ligands. In brain pericytes, PDGF-BB signaling through PDGFR-β is critical for pericyte proliferation, survival, and function, which are all important for maintaining BBB integrity. Pericytes regulate BBB integrity; loss of pericytes increases permeability. This, alongside the functional observation that PIN-alone improved neural recording quality compared to diluent controls suggests that PIN-alone may have a beneficial effect on pericytes which contribute to the BBB integrity.

Regardless, at this time and especially without the publication of our new data, we still cannot say whether this effect is due to stabilizing vasculature earlier in the time-course, producing beneficial knock-on effects, or due to subsequent dosing after implantation. Thus, we have softened the language in our results and discussion paragraphs related to PIN-Only treatment and its effects on BBB, as follows:

“Interestingly, even the PIN-only group showed moderate improvements in recording performance and reduced IgG infiltration, suggesting that the nanoparticles alone may aid BBB resealing. However, as IgG is an indirect marker and can also reflect inflammation-induced permeability, we cannot conclusively determine whether PINs directly stabilize the BBB. Future work should explore this using endothelial markers and real-time imaging.”

Discussion

- The initial portion of the Discussion repeats content that was already covered in the Introduction. This redundancy reduces the impact of the discussion and makes the manuscript unnecessarily lengthy.

We thank the authors for their observation and agreed with their assessment. We have significantly revised the discussion as part of the restructuring of the manuscript to make it more concise and readable.

- The discussion of dexamethasone side effects is appreciated, but additional data on inflammatory markers in peripheral organs (e.g., liver, spleen) would strengthen claims of systemic safety.

We agree that this data would be valuable; however, given that we have not yet finalized a dosage duration and frequency, we felt that the data presented were at least positive indicators of the safety, albeit not definitive to the level of scrutiny that is required for regulatory approval.

We have added the following to the discussion sections, dealing with potential issues during translation (also addressing Reviewer #1, Major Comment #1) (references removed here for simplicity, but remain in the main text):

“Despite these encouraging results, long-term safety remains a key consideration for the clinical translation of DEXSPPIN. Chronic glucocorticoid use is associated with well-documented side effects, including osteoporosis, hyperglycemia, and weight loss. Moreover, PINs are designed to promote hemostasis, which may carry pro-thrombotic risk during extended use. While Hickman et al. reported no thrombosis in major organs at a 1-hour timepoint post-PIN administration, further studies are needed to assess clotting risk with chronic dosing.

Translational safety is further complicated by species-specific differences in nanomedicine responses. This variability underscores the importance of testing in diverse models. Encouragingly, the base nanoparticle formulation—SynthoPlate® (Haima Therapeutics)—has already demonstrated safety and efficacy in multiple preclinical species, including mouse, rat, rabbit, and pig, with titrated effective dosing established in several studies. In our own 8-week study, no severe adverse reactions or gross behavioral changes were observed in rats, but longer-term studies in additional species will be essential for fully assessing biocompatibility and systemic effects.

Another key challenge for translation lies in the method of delivery. Although intra-arterial injection was initially chosen to avoid complications such as phlebitis from repeated tail vein access, this route is not practical in clinical settings due to the increased risk of vascular injury and embolic events, particularly when administering nanoparticle-based formulations. Future studies should investigate the durability of therapeutic effects with shorter or lower dosing regimens, explore safer and more clinically feasible delivery routes, and validate both efficacy and safety in large animal models to support translational readiness.”

Reviewer #3 (Remarks to the Author):

The authors comprehensively study the impact of systemic administration of dexamethasone loaded in platelet-inspired nanoparticles on maintaining the performance of intracortical microelectrode recording performance. The work explores nanoparticle characterization, in vitro release of dexamethasone, in vivo performance of IMEs, and potential side effects. I have suggestions that may strengthen the manuscript. Please don't be alarmed by the number of comments; most of my comments are small changes to strengthen phrasing/bring out the interesting work you all did. It would be helpful to consider the most important control groups and what each comparison offers insight into, and then draw out those conclusions for the reader.

1. First, SPPINDEX as an acronym may sound better than DEXSPPIN, but shouldn't the acronym be based on the order of the words they represent?

The acronym of “dexamethasone sodium phosphate-loaded platelet-inspired nanoparticles” was changed to DEXSPPIN throughout the entire manuscript.

2. Final sentence of abstract: add “IME” – “...that improves neuronal density and enhances IME recording performance.”

Done

Introduction:

3. First sentence of second paragraph in Introduction: is cellular resources the correct word? Cellular activities maybe? *Adjusted to cellular activities.*

4. Similar question in second paragraph: is “encapsulation” the right word? Perhaps microglia and astrocytes surround the IME interface?

“Encapsulation” was removed. The sentence below was added.

“This cascade perpetuates inflammation through damage-associated molecular pattern (DAMP) signaling, ultimately resulting in microglia and astrocytes surrounding the IME interface and further degradation of recording quality [32–36]”

5. The final sentence of the fourth paragraph of the introduction can be broken into pieces/rearranged. It's a little long and I think it has a lot of important information that would be better communicated in a concise manner.

The final sentence of the fourth paragraph was simplified as: “We hypothesized that loading PINs with dexamethasone sodium phosphate (DEXSP) would enable anti-inflammatory drug delivery to IME implant sites. This approach not only has the potential to reduce neuroinflammation but also has the potential to promote BBB resealing and hemostasis.”

6. In last paragraph of the Introduction, rearrange: “First, we investigated the feasibility of manufacturing SPPINDEX using lipid thin-film rehydration with a DEXSP-containing solution to achieve high encapsulation

efficiency.” To clarify our meaning in this section, we rephrased it as follows, using a more active voice: *“Here, we evaluate the therapeutic efficacy of systemically administered DEXSP loaded PINs (DEXSPPIN) in improving long-term IME performance. Using a lipid thin-film rehydration method, we developed a DEXSPPIN formulation that reliably achieved high drug encapsulation efficiency. We then assessed neural recording performance over 8 weeks of weekly treatment via extracellular single-unit recordings.”*

7. A question about this word choice: “...activated microglia (CD68) and astrocytes (GFAP) were **evaluated** to assess neuroinflammation.” What do you mean by “evaluated”? Can you be a little more specific? Yes, we agree that “evaluated” is a vague word in this sentence. We rephrased this sentence and added more details. *“To probe the neurobiological effects of DEXSPPIN, we quantified neuron density (NeuN), microglia activation (CD68) and astrocyte reactivity (GFAP) via immunohistochemistry around the implant site at the study endpoint. Blood-brain barrier integrity was assessed via immunoglobulin-G (IgG) staining.”*

8. The introduction suggests that the authors quantify bone mineral density. I may have missed these results. We intended to quantify these data and include them in our manuscript. Unfortunately due to the manner in which the samples were scanned using the microCT (did not include a HU calibration), and the fact that the data were acquired over numerous months (thus more likely to introduce instrumentation/calibration drift), we did not feel comfortable including these data. We have included a representative image below of one of our rat femurs for the reviewer’s interest, but do not think it adds additional value to the manuscript, given the quantitative confounds addressed above. **We have now removed the erroneous reference to BMD from the introduction.**

Figure: 3D rendering of rat femur from microCT (left) and cross-sectional view of the mid-bone (right).

Materials & Methods

9. How much time did it take for Kwik-cast to cure? It usually takes ~15 minutes to cure. This detail was added in the method section.

10. In “Dexamethasone Sodium Phosphate Loaded Nanoparticle Manufacture and Characterization”: The sentence: “Nanoparticle characterizations, including...” is phrased in a way that sounds like DLS will be used to determine hydrodynamic size, zeta potential, and polydispersity index. Rephrase for clarity. We have added more details to this section: *“Nanoparticle characterizations, including hydrodynamic size distribution, zeta potential, and polydispersity index (PDI) of DEXSPPIN, were conducted using Data Litesizer (Anton Paar, Ashland, VA). Zeta potential measurements were performed using an Omega cuvette (Mat. No. 225288, Anton Paar, Ashland, VA) and analyzed under the Smoluchowski approximation with a Henry factor of 1.5. Measurements were conducted using water as the solvent, with the refractive index set to 1.33 and relative permittivity at 78.3.”*

11. In the equation for encapsulation efficiency, can you change “total used drug” to something like “total drug” or “total drug amount”? The word “used” I associate more with the idea of something being “spent”/“finished”, and I think that can be confusing. “Total used drug” was changed to “total drug” in the equation to calculate encapsulation efficiency from method section: “Dexamethasone Sodium Phosphate Loaded Nanoparticles Purification and Encapsulation Efficiency Characterization”

12. For the different treatments for the *in vivo* study, a table could be useful.

Below treatment summary table was created and included in the method section: “Dexamethasone Sodium Phosphate Loaded Nanoparticles Administration”

Table 1. Summary of *In Vivo* Treatment Groups

Treatment	Drug Dosage (mg/kg)	Nanoparticle Dosage (mg/kg)	Frequency	N of animals
DEXSPPIN	0.3	2	Once per week	7
PIN	N/A	2	Once per week	7
DEXSP	0.3	N/A	Once per week	8
DILUENT	N/A	N/A	Once per week	7

13. What type of slides were used for brain tissue sections? The Fisherbrand Superfrost Plus microscope slides were used for collecting all brain tissue sections (now included in the methods section)

14. Did the authors use a secondary control for IHC? Standard positive and negative controls were run to validate each of the antibodies used in our study (Lam 2024). (now included in the methods section, along with a table of antibodies with additional details included in the supplementary)

15. Statistics section: list the alpha value for ANOVA. $\alpha = 0.05$ was set for all statistical analysis in this study. Alpha value was listed in method section: “Statistics”

16. Final sentence: I believe all the figures in the manuscript are bar graphs, not box plots. The typo “box plots” was changed to “bar plots” in method section: “Statistics”.

Results

17. Did the authors compare the hydrodynamic diameters of SPPINDEX and PIN?

We added the table about the characterization of PIN in supplementary (Table S2). The average hydrodynamic diameter of PIN is around 128 nm which is closer to the hydrodynamic diameter of SPPINDEX (~125 nm).

18. Figure 2 caption: list D90 in (C); isn't 5% w/v trehalose in 4.05 mg/mL HEPES what you have denoted as “TH”?

Yes, 5% w/v trehalose in 4.05 mg/mL HEPES is the buffer we used to rehydrate lipid films and diluent particles treatments. We changed acronym “TH” to “DILUENT” over the entire manuscript to improve the readability and clarity.

19. How many replicates are in Figure 2D? Where are the error bars? If first week is the most relevant (since solutions were prepared fresh weekly for the *in vivo* work), consider adding a zoomed-in version of the graph.

There are 3 parallel samples for release study in Figure 1D. The error bars were added to Figure 1D. Zoomed-in version of cumulative release curve was added (Figure 1D, up to 72 hours).

20. In “Therapeutic Efficacy of SPPINDEX on IMEs Recording Performance”: middle of the first paragraph, do not use the word “It” as it can be confusing as to what that is referring to. “It” was changed to “AEY” – active electrode yield.

21. Please reorder the Results discussion of Figure 3 or the Figure itself to match the order in which you discuss it. Both Figure 1 and Figure 2 panels were reordered to match the order in which we discuss about them.

22. I think the word “improved” should be replaced with “increased” in the second paragraph of “Therapeutic Efficacy of SPPINDEX on IMEs Recording Performance”. “Increased” is more specific and is numerical – which is important as you are talking about statistical significance here. This comment holds for other instances in the manuscript as well whenever significant differences are discussed. The reviewer is correct on the term “increased”. “Improved” terms were changed to “increased” whenever significance was discussed.

23. Figure 3:

a. Error bars on Figure 3B?

Figure 2B showing the result of active electrode yield. The definition of active electrode was described in the Methods section as follows:

“To quantify recording performance, we measured active electrode yield (AEY) biweekly, defined as the percentage of channels detecting single units (green dots in Figure 2A) relative to the total viable channels (excluding persistently silent electrodes; red dots in Figure 2A). AEY serves as a practical measure of electrode function and signal integrity over time.”

A two-sided proportion z-test was performed to compare AEY across treatment groups at each time point, as described in a previously published study by Hoferlin et al. (2025). Because the values shown in Figure 3B represent pooled proportions across all channels and animals—not mean values per animal—we were previously advised by our consulting statistician that conventional error bars are not applicable and therefore not shown.

Hoeflerlin, G. F., Grabinski, S. E., Druschel, L. N., Duncan, J. L., Burkhart, G., Weagraff, G. R., ... & Capadona, J. R. (2025). Bacteria invade the brain following intracortical microelectrode implantation, inducing gut-brain axis disruption and contributing to reduced microelectrode performance. *Nature Communications*, 16(1), 1829.

b. Why are there some rows in red in Figure 3C? These were meant to emphasize the comparisons between DEXSPPIN and TH, DEXSPPIN and Free DEXSP; To avoid confusion, we changed the color back to black throughout to avoid any confusion.

c. Mislabeled x-axes for D, F, G, H, and I. W4_8 should be **W5_8**. All changed to W5_8. In addition, Figure F, G, H, and I were moved to supplementary to address the concern from Reviewer #2 – regarding simplicity of the figure.

d. In Figure 3 caption: final sentence, unless you ran statistics comparing the decline of each group, do not use the word “significantly”. *Significantly* was removed – caption regarding Figure 3F-I were moved to supplementary materials.

24. “Noise levels in the SPPINDEX group were significantly higher compared to the TH and Free DEXSP groups ($p < 0.05$ to TH, $p < 0.001$ to Free DEXSP) during the first four weeks, indicating a potential increase in the noise associated with SPPINDEX treatment during the early weeks (Figure 4B).” The second half of this sentence is very redundant. *We removed the second half of this sentence.*

25. Why didn’t the authors comment on Figure 4D stats comparing Free DEXSP and PIN? Sometimes stats that do not include SPPINDEX are discussed, and other times they are not, and I am not sure when/why it is decided to be one way or another.

We agree with the reviewer’s comment regarding inconsistencies in the discussion of Figure 3. We have added the following text to improve clarity and ensure consistent interpretation across all panels (new text in blue): “Lastly, we examined spike rates as an indirect measure of neuronal activity. The PIN group displayed significantly lower spike rates compared to the Free DEXSP group ($p < 0.05$), while no significant differences were observed between PIN and DILUENT. These findings suggest that continuous free drug exposure may modestly enhance neuronal firing rates. Notably, the DEXSPPIN group did not differ significantly from any other treatment group in spike rate, suggesting that targeted delivery preserved baseline firing patterns without overt hyper- or hypo-activation (Figure 3D).”

26. Figure 4 caption: how did the authors choose what parts of the figure panels to discuss? (C) and (D) seem sort of odd to compare only SPPINDEX to TH. (similar comment for Figure 5E caption)

We aimed to:

- 1. Compare DEXSPPIN to all other groups to evaluate whether drug-loaded nanoparticles significantly affect recording parameters beyond the effects of the nanoparticle itself, the drug alone, and the diluent.*
- 2. Compare PIN to both the DILUENT and Free DEXSP control groups to assess whether the functionality of PIN (e.g., promoting hemostasis) impacts recording parameters independently of the drug.*

in Figure 3C caption, the follow sentence was added to describe the significance of comparison between PIN & DILUENT, PIN & Free DEXSP on SNR: “(C) Signal-to-noise ratio (SNR) was significantly lower in the DEXSPPIN group compared to the DILUENT group in both study phases. SNR was significantly lower in the PIN group compared to the DILUENT and Free DEXSP groups.”

In Figure 3D caption, we added a sentence to talk about the significance of comparison between the PIN and Free DEXSP groups: “(D) Spike rate was significantly lower in the PIN group compared to the Free DEXSP group.”

In Figure 4E caption, the following sentences were added to cover the comparison between PIN and DILUENT groups on IgG intensity: "(E) IgG intensity was significantly lower in the DEXSPPIN group compared to the DILUENT group up to 600 μm . The PIN group also shows the significant lower IgG intensity starting from 50 μm up to 600 μm from the implant site compared to the DILUENT group (see Table S3 – S6 for full statistical comparisons across different bins for different biomarkers)."

27. In "Therapeutic Effects of SPPINDEX on Neuroinflammation and BBB Permeability"

a. Final sentence of first paragraph is about microglia – can you rearrange the paragraph to discuss each panel of Figure 5 together?

We have revised this paragraph significantly to improve readability. While we chose to not discuss each portion of Figure 4 in a single paragraph, we feel the structure below may still address the reviewer's concern. The old paragraph was admittedly convoluted. We have broken this down into 3 smaller paragraphs as shown below:

"Following IME insertion, both damage-associated molecular patterns (DAMPs), such as cellular debris from injured neurons, and pathogen-associated molecular patterns (PAMPs), such as blood-derived proteins, contribute to the activation and recruitment of microglia and macrophages near the implant site [68–71]. This sustained immune activation can lead to chronic neuroinflammation, glial encapsulation, and ultimately long-term failure of IME performance.

At the 8-week endpoint, DEXSPPIN-treated animals exhibited significantly reduced microglial activation, as assessed by CD68 staining, within 50 μm of the implant site compared to the PIN, DILUENT, and Free DEXSP groups (Figure 4C). Notably, the Free DEXSP group exhibited the highest CD68 expression across all distances, indicating an exacerbated immune response despite systemic steroid delivery.

Astrocyte activation, quantified via GFAP staining, followed a similar trend. DEXSPPIN-treated animals showed significantly lower GFAP expression up to 100 μm from the implant site relative to the DILUENT and Free DEXSP groups (Figure 4D). PIN-treated animals also exhibited a modest but significant reduction in GFAP intensity up to 50 μm , suggesting a partial effect of the nanoparticle platform alone. The Free DEXSP group again showed the highest astrocyte activation, particularly within 50 μm of the implant site, aligning with elevated microglial responses in this group. Together, these findings indicate that DEXSPPIN treatment attenuates chronic neuroinflammation more effectively than free drug or vehicle controls."

b. Can the authors add commentary on comparing SPPINDEX to Free DEXSP within the BBB permeability paragraph?

Certainly, we have added the following sentences to comment on the comparison between DEXSPPIN and Free DEXSP in the BBB permeability paragraph:

"Although the DEXSPPIN and Free DEXSP groups did not differ significantly, a trend toward lower IgG levels was observed in the DEXSPPIN group. In contrast, Free DEXSP-treated animals displayed IgG intensities comparable to the DILUENT group at all distances, suggesting limited impact on BBB repair. These results support the hypothesis that targeted DEXSP delivery via the PIN platform enhances BBB resealing and reduces post-insertion microvascular permeability more effectively than free drug administration alone."

28. What do the authors think the hole is in Figure 5A SPPINDEX NeuN? It's ~northeast of the IME. If it is more of an artifact from slicing, was that portion included in the analysis?

We believe the hole observed in Figure 4A SPPINDEX NeuN is an artifact likely caused by brain cryo-slicing or the tissue fixation process. In our Python-based quantification software, we routinely exclude any regions affected by such artifacts to avoid bias in histological analysis. For example, the hole in Figure 4A DEXSPPIN NeuN and the large void near the IME site in Figure 4A DEXSPPIN IgG were also excluded from quantification to ensure accurate data interpretation.

29. “Side Effects of...” section:

a. ~middle of first paragraph: “...we measured glucose levels, alanine transaminase (ALT) levels, and creatinine (CREA) levels at week 0, week 4, and week 8, weights weekly.” What is meant by “weights weekly”? I assume it means the authors weighed the rats weekly. Right now, “weights weekly” is not a complete thought. Perhaps: “...at week 0, 4, and 8, we measured glucose levels, alanine transaminase (ALT) levels, rat weights, and creatinine (CREA) levels.” (please make sure figure panel order and the order each is discussed matches after these changes)

The weights of rats were measured weekly. The blood parameters of rats were measured at week 0, week 4, and week 8. The unclarity in the manuscript was revised as below, which is now consistent with the presentation order in Figure 6:

“Body weights were recorded weekly, and glucose, ALT, and CREA levels were assessed at baseline (week 0), midpoint (week 4), and study endpoint (week 8).”

b. It is unnecessary to write “..., and therefore the data was omitted from plotting in Figure 6.” It makes sense why you would include it only in supplementary.

The sentence was removed from the manuscript.

Discussion

30. Overall, the discussion is great. Some of the organization can be adjusted to keep similar topics together (e.g., second, eighth, and ninth paragraphs).

We have substantially revised the discussion section to address this comment and hope that the reviewer agrees it has strengthened the organization and clarity.

31. I find some of the cytokine and pathway discussion to be excessive for something that was unfortunately unable to be captured for this study. I do appreciate that it is mentioned but feel it could be briefer.

We have substantially shortened the paragraph discussion the limitation of the study regarding cytokines and molecular pathways, as follows.

“Mechanistically, we propose that DEXSP released from PINs binds to glucocorticoid receptors on activated microglia, suppressing NF- κ B signaling through I κ B α upregulation [86–88]. This may disrupt the self-sustaining inflammatory loop at the implant site, reducing cytokine production, limiting astrocyte recruitment, and preventing secondary neuron death. Although we were unable to directly quantify proinflammatory cytokines due to technical limitations with fixed tissue, future studies could apply spatial transcriptomics or NanoString analysis to explore local gene expression profiles [89].”

32. The middle of the fourth paragraph claims the “principal hypothesis of the current study is that systemic administration of PIN can promote hemostasis at the site of insertion-induced microhemorrhages and reseal the compromised BBB.” This does not make it seem like SPPINDEX is the primary focus of the work. If microhemorrhages are a focus, were characteristics of the TH control group, or free DEXSP for that matter, consistent with microhemorrhages? If that is the case, bring that out more.

We appreciate the feedback and have revised the paragraph as below:

“In this study, we evaluated a novel strategy combining injury-targeted delivery and hemostatic activity via platelet-inspired nanoparticles (PINs) loaded with dexamethasone sodium phosphate (DEXSP). We hypothesized that systemic DEXSPPIN would enhance drug localization at the implant site, promote BBB resealing, and synergistically attenuate neuroinflammation to improve IME recording performance.”

33. Change the reference to citation 87 to **Song et al.** rather than Sydney. Revised to Song et al.

34. At times, it is unclear which control groups are the most important for this work. Ultimately, do the authors recommend systemic administration of SPPINDEX or PIN?

We have clarified our principal hypothesis in response to comment #32, and further strengthened this point in our conclusion as shown below:

“This study demonstrates that targeted delivery of DEXSP via platelet-inspired nanoparticles significantly improves the long-term recording performance of intracortical microelectrodes. By combining injury-targeted drug delivery with hemostatic activity, DEXSPPIN mitigates neuroinflammation, preserves neuron density, and reduces BBB permeability near the implant site. In contrast, non-targeted Free DEXSP worsened inflammation and signal quality, emphasizing the need for localized therapeutic approaches. While the PIN group alone demonstrated some benefit in recording performance over the DILUENT control, it was modest compared to the DEX-loaded DEXSPPIN group. Future studies should focus on refining dosing strategies, characterizing molecular mechanisms, and evaluating long-term safety to advance the clinical translation of this promising neuroprotective strategy.”

35. Consider ending with the most important comparisons and positive takeaways from the work. Although discussing the translatability is important, ending the discussion section with it detracts from interesting, helpful findings from the actual work.

We have taken the reviewer’s advice, and revised the conclusion (see #34 above).

Conclusion

36. Since pathways were not specifically explored (and is later mentioned as a future idea in the same paragraph), I think it is a reach to say that SPPINDEX disrupted signaling pathways. Instead, it may be important to mention how SPPINDEX compares to free DEXSP.

We have removed conjecture regarding the signaling pathways from the conclusion, and revised as discussed in the response to #34 and #35.

Small comments & suggestions (mostly grammatical) that’ll elevate the work even more:

- Please make sure you mention/call to all figures in the manuscript. Figures 2C, 3A, and 5A are not mentioned in the text. – **They are now all mentioned in the manuscript.**

- Acronyms: please go through and check the first instance an acronym is mentioned and delete all redundant defining of those acronyms (e.g., BBB is defined twice in the abstract; SPPINDEX is defined numerous times throughout the manuscript); please define other acronyms (e.g., TNF and IL); DAMPs and PAMPs should be defined as well – I am not sure what the phrases in the parentheses are for.

All acronyms throughout the manuscript were reviewed and revised. Acronyms are defined upon first use within the main text and within figure captions per the journal guidelines. Redundant definitions within the same section were removed. Undefined acronyms, including TNF, IL, DAMPs, and PAMPs, have now been properly defined.

- Go through and check verb tenses, especially through the results section. An example is the caption for Figure 4 – (A) is in present tense, but the other parts are in past tense. Another in the introduction: in PIN section, “Therefore, we hypothesize that PIN can be used to targeted deliver anti-inflammatory...” **We have thoroughly proofread the manuscript and revised verb tenses throughout to ensure consistency and accuracy.**

- A few instances of extra uses of the word: “the”. For example, in the abstract: “Recent studies suggest that the leakage of blood-brain barrier (BBB) and microhemorrhage caused by **the** IME insertions lead to **the** increased neuroinflammation and reduced neural...”. There are also a few places I felt it was missing (e.g., second paragraph of introduction: “Upon the insertion of IMEs into **the** brain motor cortex,...”). **We reviewed the entire manuscript to remove redundant uses of 'the' and to add missing instances where appropriate.**

- A few Oxford commas are missing – a personal preference but sometimes it certainly would aid in increasing reader understanding. Missing oxford commas were added.
- Remove “ing” from the last sentence of the third paragraph of the introduction: “...few proposed therapies specifically targeting the initial prevention of foreign...”. Removed.
- A few periods are missing. One in the methods section “Electrophysiological Analysis” – near the end of the first paragraph “...time **phase. On** a weekly basis,...”. Missing periods were added.
- Another small change is to be consistent with the order that experiments are listed and then presented in the text later (e.g., the last sentence of the introduction lists some metrics that are tracked/evaluated in a different order than the results are presented later). Reviewed and revised.
- Please move “(Sprague Dawley...)” after “29 male rats” instead of after “catheters” in Methods section. (Sprague Dawley, CD, Charles River Labs, Wilmington, MA) was moved after “29 male rats”
- When referencing Song et al. in the discussion, I believe the authors meant DNA/RNA. DNA/RNA-related content when referring to Song et al was removed. See also in response to comment #31.

The authors comprehensively study the impact of systemic administration of dexamethasone loaded in platelet-inspired nanoparticles on maintaining the performance of intracortical microelectrode recording performance. The work explores nanoparticle characterization, in vitro release of dexamethasone, in vivo performance of IMEs, and potential side effects. I have suggestions that may strengthen the manuscript. Please don't be alarmed by the number of comments; most of my comments are small changes to strengthen phrasing/bring out the interesting work you all did. It would be helpful to consider the most important control groups and what each comparison offers insight into, and then draw out those conclusions for the reader.

1. First, SPPINDEX as an acronym may sound better than DEXSPPIN, but shouldn't the acronym be based on the order of the words they represent?
2. Final sentence of abstract: add "IME" – "...that improves neuronal density and enhances **IME** recording performance."

Introduction:

3. First sentence of second paragraph in Introduction: is cellular *resources* the correct word? Cellular *activities* maybe?
4. Similar question in second paragraph: is "encapsulation" the right word? Perhaps microglia and astrocytes *surround* the IME interface?
5. The final sentence of the fourth paragraph of the introduction can be broken into pieces/rearranged. It's a little long and I think it has a lot of important information that would be better communicated in a concise manner.
6. In last paragraph of the Introduction, rearrange: "First, we investigated the feasibility of manufacturing SPPINDEX using lipid thin-film rehydration with a DEXSP-containing solution to achieve high encapsulation efficiency."
7. A question about this word choice: "...activated microglia (CD68) and astrocytes (GFAP) were **evaluated** to assess neuroinflammation." What do you mean by "evaluated"? Can you be a little more specific?
8. The introduction suggests that the authors quantify bone mineral density. I may have missed these results.

Materials & Methods

9. How much time did it take for Kwik-cast to cure?
10. In "Dexamethasone Sodium Phosphate Loaded Nanoparticle Manufacture and Characterization": The sentence: "Nanoparticle characterizations, including..." is phrased in a way that sounds like DLS will be used to determine hydrodynamic size, zeta potential, and polydispersity index. Rephrase for clarity.
11. In the equation for encapsulation efficiency, can you change "total used drug" to something like "total drug" or "total drug amount"? The word "used" I associate more with the idea of something being "spent"/"finished", and I think that can be confusing.
12. For the different treatments for the *in vivo* study, a table could be useful.
13. What type of slides were used for brain tissue sections?
14. Did the authors use a secondary control for IHC?
15. Statistics section: list the alpha value for ANOVA.
16. Final sentence: I believe all the figures in the manuscript are bar graphs, not box plots.

Results

17. Did the authors compare the hydrodynamic diameters of SPPINDEX and PIN?

18. Figure 2 caption: list D90 in (C); isn't 5% w/v trehalose in 4.05 mg/mL HEPES what you have denoted as "TH"?
19. How many replicates are in Figure 2D? Where are the error bars? If first week is the most relevant (since solutions were prepared fresh weekly for the *in vivo* work), consider adding a zoomed-in version of the graph.
20. In "Therapeutic Efficacy of SPPINDEX on IMEs Recording Performance": middle of the first paragraph, do not use the word "It" as it can be confusing as to what that is referring to.
21. Please reorder the Results discussion of Figure 3 or the Figure itself to match the order in which you discuss it.
22. I think the word "improved" should be replaced with "increased" in the second paragraph of "Therapeutic Efficacy of SPPINDEX on IMEs Recording Performance". "Increased" is more specific and is numerical – which is important as you are talking about statistical significance here. This comment holds for other instances in the manuscript as well whenever significant differences are discussed.
23. Figure 3:
 - a. Error bars on Figure 3B?
 - b. Why are there some rows in red in Figure 3C?
 - c. Mislabeled x-axes for D, F, G, H, and I. W4_8 should be **W5_8**.
 - d. In Figure 3 caption: final sentence, unless you ran statistics comparing the decline of each group, do not use the word "significantly".
24. "Noise levels in the SPPINDEX group were significantly higher compared to the TH and Free DEXSP groups ($p < 0.05$ to TH, $p < 0.001$ to Free DEXSP) during the first four weeks, indicating a potential increase in the noise associated with SPPINDEX treatment during the early weeks (Figure 4B)." The second half of this sentence is very redundant.
25. Why didn't the authors comment on Figure 4D stats comparing Free DEXSP and PIN? Sometimes stats that do not include SPPINDEX are discussed, and other times they are not, and I am not sure when/why it is decided to be one way or another.
26. Figure 4 caption: how did the authors choose what parts of the figure panels to discuss? (C) and (D) seem sort of odd to compare only SPPINDEX to TH. (similar comment for Figure 5E caption)
27. In "Therapeutic Effects of SPPINDEX on Neuroinflammation and BBB Permeability"
 - a. Final sentence of first paragraph is about microglia – can you rearrange the paragraph to discuss each panel of Figure 5 together?
 - b. Can the authors add commentary on comparing SPPINDEX to Free DEXSP within the BBB permeability paragraph?
28. What do the authors think the hole is in Figure 5A SPPINDEX NeuN? It's ~northeast of the IME. If it is more of an artifact from slicing, was that portion included in the analysis?
29. "Side Effects of..." section:
 - a. ~middle of first paragraph: "...we measured glucose levels, alanine transaminase (ALT) levels, and creatinine (CREA) levels at week 0, week 4, and week 8, weights weekly." What is meant by "weights weekly"? I assume it means the authors weighed the rats weekly. Right now, "weights weekly" is not a complete thought. Perhaps: "...at week 0, 4, and 8, we measured glucose levels, alanine transaminase (ALT) levels, rat weights, and creatinine (CREA) levels." (please make sure figure panel order and the order each is discussed matches after these changes)
 - b. It is unnecessary to write "..., and therefore the data was omitted from plotting in Figure 6." It makes sense why you would include it only in supplementary.

Discussion

30. Overall, the discussion is great. Some of the organization can be adjusted to keep similar topics together (e.g., second, eighth, and ninth paragraphs).
31. I find some of the cytokine and pathway discussion to be excessive for something that was unfortunately unable to be captured for this study. I do appreciate that it is mentioned but feel it could be briefer.
32. The middle of the fourth paragraph claims the “principal hypothesis of the current study is that systemic administration of PIN can promote hemostasis at the site of insertion-induced microhemorrhages and reseal the compromised BBB.” This does not make it seem like SPPINDEX is the primary focus of the work. If microhemorrhages are a focus, were characteristics of the TH control group, or free DEXSP for that matter, consistent with microhemorrhages? If that is the case, bring that out more.
33. Change the reference to citation 87 to **Song et al.** rather than Sydney.
34. At times, it is unclear which control groups are the most important for this work. Ultimately, do the authors recommend systemic administration of SPPINDEX or PIN?
35. Consider ending with the most important comparisons and positive takeaways from the work. Although discussing the translatability is important, ending the discussion section with it detracts from interesting, helpful findings from the actual work.

Conclusion

36. Since pathways were not specifically explored (and is later mentioned as a future idea in the same paragraph), I think it is a reach to say that SPPINDEX disrupted signaling pathways. Instead, it may be important to mention how SPPINDEX compares to free DEXSP.

Small comments & suggestions (mostly grammatical) that'll elevate the work even more:

- Please make sure you mention/call to all figures in the manuscript. Figures 2C, 3A, and 5A are not mentioned in the text.
- Acronyms: please go through and check the first instance an acronym is mentioned and delete all redundant defining of those acronyms (e.g., BBB is defined twice in the abstract; SPPINDEX is defined numerous times throughout the manuscript); please define other acronyms (e.g., TNF and IL); DAMPs and PAMPs should be defined as well – I am not sure what the phrases in the parentheses are for.
- Go through and check verb tenses, especially through the results section. An example is the caption for Figure 4 – (A) is in present tense, but the other parts are in past tense. Another in the introduction: in PIN section, “Therefore, we hypothesize that PIN can be used to targeted deliver anti-inflammatory...”
- A few instances of extra uses of the word: “the”. For example, in the abstract: “Recent studies suggest that the leakage of blood-brain barrier (BBB) and microhemorrhage caused by **the** IME insertions lead to **the** increased neuroinflammation and reduced neural...”. There are also a few places I felt it was missing (e.g., second paragraph of introduction: “Upon the insertion of IMEs into **the** brain motor cortex,...”).
- A few Oxford commas are missing – a personal preference but sometimes it certainly would aid in increasing reader understanding.
- Remove “ing” from the last sentence of the third paragraph of the introduction: “...few proposed therapies specifically targeting the initial prevention of foreign...”.
- A few periods are missing. One in the methods section “Electrophysiological Analysis” – near the end of the first paragraph “...time **phase. On** a weekly basis,...”.
- Another small change is to be consistent with the order that experiments are listed and then presented in the text later (e.g., the last sentence of the introduction lists some

metrics that are tracked/evaluated in a different order than the results are presented later).

- Please move "(Sprague Dawley...)" after "29 male rats" instead of after "catheters" in Methods section.
- When referencing Song et al. in the discussion, I believe the authors meant DNA/RNA.